# The Discretization Complexity Analysis of Consistency Models under Variance Exploding Forward Process

## Abstract

Consistency models, a new class of one-step generative models, have shown state-of-the-art performance in one-step generation and achieve competitive performance compared to multi-step diffusion models. The most challenging part of consistency models is the training process, which discretizes the diffusion process and trains a consistency function to map any point at any discretized timepoint of the diffusion process to the data distribution. Despite the empirical success, only a few works focus on the discretization complexity of consistency models. However, the setting of those works is far away from the empirical consistency models with good performance, suffers from large discretization complexity, and fails to explain the empirical success of consistency models. To bridge the gap between theory and application, we analyze consistency models with two key properties: (1) variance exploding forward process and (2) gradually decay discretization stepsize, which are both widely used in empirical consistency models. Under the above realistic setting, we make the first step to explain the empirical success of consistency models and achieve the state-of-the-art discretization complexity for consistency models, which is competitive with the results of diffusion models. After obtaining the results of the one-step sampling method of consistency models, we further analyze a multi-step consistency sampling algorithm proposed by Song et al. (2023) and show that this algorithm improves the discretization complexity compared with one-step generation, which matches the empirical observation.

## 1 Introduction

Recently, diffusion models have shown impressive performance in different areas such as image generation and video generation (Rombach et al., 2022; Esser et al., 2024; Ho et al., 2022). The mathematical mechanism of diffusion models is made up of two processes: the forward process and the reverse process. The forward process gradually injects noise into data till the marginal distribution is close to pure noise. The reverse process is an iterative sampling process, which sequentially removes noise from data to generate clean samples. At each denoised step, diffusion models only need to predict and remove a small noise, making the training process more stable than Generative Adversarial Networks (GAN) (Goodfellow et al., 2014). However, the iterative sampling process indicates diffusion models need to evaluate a large neural network to predict the noise at this step, leading to a higher computational cost than other one-step algorithms such as GAN, Variational Auto-Encoder (Kingma and Welling, 2013), and Normalizing Flow (Papamakarios et al., 2021).

To solve the computational issue, a series of works try to accelerate the sampling process of diffusion models (Song et al., 2020; Bao et al., 2022; Zheng et al., 2023). One notable algorithm in this series work is the Consistency Model (Song et al., 2023), which tries to find a mapping function (a.k.a. consistency function) to directly map any points at any time of the forward process to the data distribution. Consistency models have shown state-of-the-art (SOTA) performance compared to other one-step generative models in image generation (Song et al., 2023; Kim et al., 2023), video generation (Wang et al., 2023) and music generation (Fei et al., 2024). Furthermore, it is also widely used in other areas, such as reinforcement learning (Ding and Jin, 2023).

To obtain the consistency function, consistency models discretize the timeline into $K$ discretization points and hope the consistency function outputs similar results at two adjacent discretization points in the training phase. When the two adjacent points are far away, it is hard to train a consistency function with great performance, which indicates $K$ can not be too small. However, it is also necessary to avoid a large $K$ since it will make the training phase time-consuming. Hence, one core of consistency models training phase is the choice of the discretization number $K$, which is helpul in effective training a consistency function with great performance.

Despite the empirical success of consistency models, no existing works explain why consistency models achieve comparable performance to diffusion models. Though some impressive works analyze the discretization complexity of consistency models (Lyu et al., 2024; Li et al., 2024; Dou et al.), the setting is far away from the consistency models with great performance in application:

- **The forward process.** Previous theoretical works choose the variance preserving forward process (VPSDE) to inject noise in the training phase. On the contrary, empirical consistency models adopt the variance exploding forward process (VESDE) with a specific noise schedule whose solution trajectory is linear. As discussed in Karras et al. (2022) and Liu et al. (2022), a linear trajectory indicates it is possible to generate clean samples with a single Euler step and is the basis of a one-step model.
- **The discretization scheme.** The discretization scheme of consistency models is the EDM scheme (Eq. (5)), which first uses a large stepsize and gradually reduces the stepsize. However, Lyu et al. (2024) and Dou et al. use a uniform stepsize, and Li et al. (2024) use a scheme that relies heavily on the VPSDE (Detail in Appendix C.1).

Due to the mismatch between the current theoretical and empirical setting, the discretization complexity results of current theoretical works are significantly worse than diffusion models (Table 1). Hence, the following natural problem remains open:

*Under a realistic setting with great empirical performance, is it possible to achieve complexity comparable to the state-of-the-art diffusion models and explain the success of consistency models?*

In this work, for the first time, we analyze consistency models under the VESDE forward process and EDM stepsize and achieve the following results.

**Theorem 1.** (Informal) *Assume bounded supported target data distribution. With suitable assumption on pre-trained score function, consistency function [1], the consistency models require*

$$K = O\left( \frac{R^{4+\frac{3}{2a}} d^{3+\frac{1}{2a}}}{\epsilon_{W_2}^{4+\frac{3}{2a}}} \right)$$

*discretization steps in the training phase to output a distribution which is Wasserstein-2 ($W_2$) close to the target distribution in the sampling phase, where $R$ is the diameter of target distribution $X_0 \sim q_0 \in \mathbb{R}^d$ and $a = 7$ is a parameter of EDM (5) stepsize.*

As shown in Table 1, this result is better than the previous theoretical guarantee of consistency models, shows the benefit of a suitable $a$ and achieves competitive results compared to SOTA diffusion model results. The core step of this result is to make full use of the time-dependence Lipschitz constant of the consistency function. More specifically, we first show that with a uniform Lipschitz constant, we achieve $1/\epsilon_{W_2}^7$ discretization complexity for any $a$. This result indicates that a uniform Lipschitz constant prevents an improved $K$ with a suitable $a$, which does not match the empirical observation (Karras et al., 2022). Then, we prove that with a time-dependent Lipschitz constant, the influence of $a$ is highlighted, and we achieve SOTA complexity for consistency models (More detail in Section 4.2).

To improve the sampling quality of consistency models, Song et al. (2023) further provide a multi-step sampling algorithm. Let $T$ be the forward diffusion time, $N$ be the number of sampling steps and $T = \tau_1 \geq \tau_2 \geq \ldots \geq \tau_N$ be a sequence of time points used in a multi-step sampling algorithm. The algorithm adds noise to the latest generated clean samples and maps the noised sample to the clean samples using the consistency function (Eq. (6)). The great performance indicates that the requirement of $K$ can be relaxed if the multi-step sampling method is used. Recently, Lyu et al.

---

[1]Different from a uniform Lipschitz constant used in previous theoretical works, we assume this Lipschitz constant depends on noise level, which matches the true order and is more realistic.

Table 1: The discretization complexity for consistency and diffusion models with reverse PFODE in Wasserstein distance. To make a thorough comparison, we also provide the SOTA discretization results for diffusion models with reverse SDE. (+) means that we transform the log-concave distribution, which is unrealistic and ignore the blow-up phenomenon of the score, to our bounded support distribution. (*) means that we transform the results of previous works into the results under our setting. We present more detail in Appendix C.1.

| Model | Forward Process | Stepsize | Complexity | Reference |
|---|---|---|---|---|
| Diffusion | VESDE | Uniform | $1/\epsilon_{W_2}^4$ 
 $1/(\epsilon_{W_2}^4 \text{Poly}(\epsilon_{W_2}))$ (+) | Gao and Zhu (2024) |
| | VPSDE | Uniform | $1/\epsilon_{W_2}$ 
 $1/(\epsilon_{W_2}\text{Poly}(\epsilon_{W_2}))$ (+) | |
| | VPSDE (Reverse SDE) | Exponential Decay | $1/\epsilon_{W_2}^4$ | Chen et al. (2022a) |
| Consistency | VPSDE | Uniform | $L_f/\epsilon_{W_2}^7$ 
 $1/\epsilon_{W_2}^9$ (*) | Lyu et al. (2024) |
| | | Specific to VPSDE | $L_f^3/\epsilon_{W_1}$ 
 $1/(\epsilon_{W_1}\epsilon_{W_2}^6)$ (*) | Li et al. (2024) |
| | | Uniform | $L_f^2 L_{\text{score}}^2/\epsilon_{W_1}^2$ 
 $1/(\epsilon_{W_1}^2\epsilon_{W_2}^{12})$ (*) | Dou et al. |
| | VESDE | EDM (5) $a \in [1,\infty)$ | $1/\epsilon_{W_2}^{4+\frac{3}{2a}}$ | Theorem 1 |
| | | | $1/\epsilon_{W_2}^{4+\frac{5}{4a}}$ | Corollary 2 (Case 2) (Multi-step Sampling) |
| | | Exponential Decay | $1/\epsilon_{W_2}^4$ | Corollary 1 |

(2024) make the first step to analyze the benefit of the multi-step sampling algorithm and show that this operation can reduce the $W_2$ error. However, since they adopt the VPSDE process, the benefit of the $W_2$ guarantee does not lead to an improved discretization complexity $K$. In this work, we show that with a VESDE process and a suitable $\{\tau_n\}_{n=1}^N$, the multi-step sampling algorithm can effectively reduce the requirement of $K$. Since $N = 2$ is enough to improve the generation quality (Song et al., 2023), we adopt this choice, achieve $O(1/\epsilon_{W_2}^{4+5/4a})$ discretization complexity and also reduce the requirement of the approximated consistency function (detail in Section 4.1). The above result is better than results in Theorem 1 and explains the role of multi-step sampling.

In conclusion, we explain why consistency models have competitive performance compared to diffusion models in application. More specifically,

- We bridge the gap between theoretical analysis and real-world applied models by analyzing consistency models with the VESDE forward process and EDM discretization scheme.

- Under the above realistic setting, we achieve the state-of-the-art discretization complexity for consistency models, which is also competitive with the results of diffusion models.

- For the first time, we also show that the multi-step sampling algorithm of consistency models can effectively reduce the discretization complexity and the requirement of other error terms.

## 2 RELATED WORK

Since the mathematical mechanism of consistency models is close to diffusion models, we discuss the discretization complexity of diffusion and consistency models. For diffusion models, we summarize the results for reverse PFODE due to the deterministic sample process of consistency models.

**Diffusion models with reverse PFODE.** With strong assumptions or additional components, a series of works achieve polynomial complexity for reverse PFODE (Li et al., 2023; Chen et al., 2023c; Gao and Zhu, 2024). More specifically, Li et al. (2023) assume an accurate enough Jacobian matrix, and Gao and Zhu (2024) assume the target data distribution is log-concave. We note that the log-concave assumption on data distribution is much stronger than our bounded support assumption since it

precludes the existence of multi-modal real-world data. Furthermore, under log-concave distribution, the ground truth score function $\nabla \log q_{T-t}(\cdot)$ does not blow-up, which can not explain the blow-up phenomenon of score function [2] and ignore the influence of early stopping parameter $\delta$ (Table 1). Without any strong assumption, Chen et al. (2023c) introduce a predictor-corrector algorithm, which switches between a Langevin corrector and a PFODE predictor. Then, they prove a polynomial discretization complexity for this algorithm.

**Consistency models.** Lyu et al. (2024) and Li et al. (2024) analyze the discretization complexity of consistency distillation and consistency training paradigm, respectively. More recently, Dou et al. analyze the estimation error and discretization complexity of consistency distillation and training paradigm at the same time. Though these works deepen the understanding of consistency models, the setting of these works is far away from consistency models in the application and suffers from large discretization complexity compared to the state-of-the-art results of diffusion models (Table 1).

## 3 THE INTRODUCTION OF DIFFUSION AND CONSISTENCY MODELS

Since the training phase of consistency models relies heavily on the diffusion process, we first introduce some basic knowledge of diffusion models. After that, we introduce how to train a consistency model with a pre-trained diffusion model.

### 3.1 THE DIFFUSION MODELS

Diffusion models consist of two processes: the forward and reverse process. The forward process converts data distribution to pure noise by adding Gaussian noise step by step. To generate clean samples, diffusion models reverse the forward process and run the corresponding reverse process.

**The forward process.** Let $q_0$ denote the data distribution. The goal of the forward process is to gradually convert $q_0$ to Gaussian noise. Given $X_0 \in \mathbb{R}^d \sim q_0$, the general forward process is

$$\mathrm{d}X_t = f(X_t, t)\,\mathrm{d}t + g(t)\,\mathrm{d}B_t, \quad X_0 \sim q_0\,,$$

where $(B_t)_{t \geq 0}$ is a $d$-dimensional Brownian motion, $f(X_t, t)$ is a drift coefficient, and $g(t)$ is a diffusion coefficient. Let $q_t$ be the density function of the forward process at time $t$ and $\{\beta_t\}_{t \in [0,T]}$ be a non-negative non-decreasing sequence. When $f(X_t, t) = -\beta_t X_t$ and $g(t) = \sqrt{2\beta_t}$, the general forward process is instantiated as a widely used variance preserving forward process (VPSDE) (Ho et al., 2020). We note that though VPSDE plays an important role in developing diffusion models, the solution trajectory of VPSDE is curved instead of linear, which prevents it from becoming the basis of one-step generative models. Hence, we focus on the variance exploding forward process, which has a linear solution trajectory under a specific noise schedule and has been widely used in many areas such as image generation (Karras et al., 2022), one-step generation (Kim et al., 2023), video generation (Blattmann et al., 2023) and intrinsic dimension estimator (Stanczuk et al.).

Let $\{\sigma_t^2\}_{t \in [0,T]}$ a non-decreasing sequence and $g(t) = \sqrt{\mathrm{d}\sigma_t^2/\mathrm{d}t}$. Then, VESDE is defined by:

$$\mathrm{d}X_t = g(t)\,\mathrm{d}B_t, \quad X_0 \sim q_0\,. \tag{1}$$

As shown in Karras et al. (2022), when choosing $\sigma_t^2 = t^2$, the solution trajectory of VESDE is linear. Hence, consistency models adopt VESDE ($\sigma_t^2 = t^2$) as the forward process. In this work, we also choose VESDE with $\sigma_t^2 = t^2$ as the forward process to match the empirical setting.

**The reverse process.** Let $t' = T - t$ be the reverse time and $(Y_{t'})_{t' \in [0,T]} = (X_{T-t'})_{t' \in [0,T]}$. To generate samples, the model reverses the forward process (Eq. (1)) and obtains the reverse process:

$$\mathrm{d}Y_{t'} = \frac{1+\eta^2}{2} g(T-t')^2 \nabla \log q_{T-t'}(Y_{t'})\,\mathrm{d}t' + \eta g(T-t')\,\mathrm{d}B_{t'}, Y_0 \sim q_T\,, \tag{2}$$

where $\eta \in [0,1]$ determines the type of the reverse process and consistency models adopt the reverse probability flow ODE (PFODE, $\eta = 0$) in its training phase. Since the ground truth score function $\nabla \log q_t(\cdot)$ and $q_T$ contain the data information, we can not directly run the above PFODE to generate

---

[2]The blow-up phenomenon means the ground truth score function $\nabla \log q_{T-t}(\cdot)$ goes to $+\infty$ at the end of the reverse process. As shown in De Bortoli (2022), **Assumption 1** allows this phenomenon.

samples. For the reverse beginning distribution $q_T$, we choose $\mathcal{N}(0, T^2 I_d)$ to approximate it due to $\sigma_T^2 = T^2$. For $\nabla \log q_t(\cdot)$, Vincent (2011) propose the following score matching objective function to learn an approximated score function $s_\phi(X_t, t)$, $\forall t \in [0, T]$:

$$\min_{\phi \in \Phi} \int_0^T w(t) \mathbb{E}_{X_0} \left[ \mathbb{E}_{X_t | X_0} \| \nabla \log q_t(X_t | X_0) - s_\phi(X_t, t) \|_2^2 \right] \mathrm{d}t \,,$$

where $w(t)$ is a weight function. With the approximated score function $s_\phi(\cdot)$, diffusion models discretize the reverse process and generate samples. Let $t_0 \leq t_1 \leq \cdots \leq t_K = T$ be the discretization points in the forward time and $h_k := t_k - t_{k-1}$ be the stepsize. When considering the reverse process, we define by $t_k' = T - t_{K-k}$ and $h_k' = h_{K-k}$ the discretization points and stepsize in the reverse process. Since the score function $\nabla \log q_{T-t}$ goes to $+\infty$ at the end of the reverse process, we adapt the early stopping technique by setting $t_0 = \delta$ to avoid this issue, which is widely used in the application (Kim et al., 2021). Then, starting from $\bar{Y}_0 \sim \mathcal{N}(0, T^2 I_d)$, we run the following process in each interval $t \in [t_k', t_{k+1}'], k \in [0, K-1]$ to generate samples:

$$\mathrm{d}\bar{Y}_{t'} = \frac{g(T - t')^2}{2} s_\phi \left( \bar{Y}_{t_k'}, T - t_k' \right) \mathrm{d}t', \quad t' \in [t_k', t_{k+1}']. \tag{3}$$

### 3.2 THE CONSISTENCY MODELS

There are two paradigms for the training phase of consistency models: consistency distillation and consistency training, where the consistency distillation paradigm requires a pre-trained score function $s_\phi(X, t)$, $\forall t \in [0, T]$ and consistency training paradigm trains in isolation. Since consistency training can not take information from a pre-trained score function, its hyperparameters need to be carefully selected to achieve great performance (Song et al., 2023; Song and Dhariwal, 2023). Hence, we analyze the discretization complexity of the consistency distillation paradigm in this work. We also discuss the current results of the consistency training paradigm in Remark 1.

Let $\boldsymbol{v}^{\mathrm{ex}}(Y, t') = \frac{g(T - t')^2}{2} \nabla \log q_{T-t'}(Y)$ be the exact vector field of PFODE (Eq. (2), $\eta = 0$). We can define the associate backward mapping $\boldsymbol{f}^{\boldsymbol{v}^{\mathrm{ex}}} : \mathbb{R}^d \times \mathbb{R}^+ \to \mathbb{R}^d$ such that for any $t' \in [0, T - \delta]$.

$$\boldsymbol{f}^{\boldsymbol{v}^{\mathrm{ex}}} (Y_{t'}, t') = Y_{T-\delta} = X_\delta \,,$$

where $\delta$ is the early stopping parameter. The above equation is equivalent to the following conditions:

$$\boldsymbol{f}^{\boldsymbol{v}^{\mathrm{ex}}} (Y_{t'}, t') = \boldsymbol{f}^{\boldsymbol{v}^{\mathrm{ex}}} (Y_{t^{\circ\prime}}, t^{\circ\prime}), \forall 0 \leq t^{\circ\prime}, t' \leq T - \delta, \text{ and}$$

$$\boldsymbol{f}^{\boldsymbol{v}^{\mathrm{ex}}} (Y, T - \delta) = Y, \, \forall Y \in \mathbb{R}^d$$

We also define the empirical vector field $\boldsymbol{v}^{\mathrm{em}}(Y, t') = \frac{g(T - t')^2}{2} s_\phi(Y, T - t')$ and the corresponding empirical backward mapping function $\boldsymbol{f}^{\boldsymbol{v}^{\mathrm{em}}}$. The goal of consistency model is to learn a consistency function $\boldsymbol{f}_{\boldsymbol{\theta}}$ to approximate $\boldsymbol{f}^{\mathrm{ex}}$ (for simplicity, we abbreviate $\boldsymbol{f}^{\boldsymbol{v}^{\mathrm{ex}}}$ as $\boldsymbol{f}^{\mathrm{ex}}$). Let $F_{\boldsymbol{\theta}}(Y, t')$ be a free-form deep neural network whose output has the same dimensionality as $Y$. To satisfy the boundary condition, (Song et al., 2023) use a skip connection:

$$\boldsymbol{f}_{\boldsymbol{\theta}}(Y, t') = c_{\mathrm{skip}}(t') Y + c_{\mathrm{out}}(t') F_{\boldsymbol{\theta}}(Y, t') \,,$$

where $c_{\mathrm{skip}}(t)$ and $c_{\mathrm{out}}(t)$ are differentiable functions such that $c_{\mathrm{skip}}(T - \delta) = 1$, and $c_{\mathrm{out}}(T - \delta) = 0$. After that, we define the consistency distillation objective function:

$$\mathcal{L}_{\mathrm{CD}}^K \left( \boldsymbol{\theta}, \boldsymbol{\theta}^-; \boldsymbol{\phi} \right) := \mathbb{E}_{X_0} \left[ \mathbb{E}_{Y_{t_k'} | X_0} \lambda \left( t_{k+1}' \right) \left\| \boldsymbol{f}_{\boldsymbol{\theta}} \left( Y_{t_k'}, t_k' \right) - \boldsymbol{f}_{\boldsymbol{\theta}^-} \left( \hat{Y}_{t_{k+1}'}^{\boldsymbol{\phi}}, t_{k+1}' \right) \right\|_2^2 \right], \tag{4}$$

where $t_k'$ is the time discretization points in the reverse process. Since $Y_{t_k'}$ is equal to $X_{T-t_k'}$, we can calculate $Y_{t_k'} | X_0$ by the forward process $X_0 + (T - t_k')Z$, where $Z$ is the standard gaussian noise. To make the training process more stable, Song et al. (2023) take similar idea with contrastive learning, stop the gradient of $\boldsymbol{\theta}^-$ and use an exponential moving average (EMA) strategy to update it $\boldsymbol{\theta}^- = \mathrm{stopgrad}\left( \mu \boldsymbol{\theta}^- + (1 - \mu)\boldsymbol{\theta} \right)$, where $\mu$ is the decay rate. For the $\hat{Y}_{t_{k+1}'}^{\boldsymbol{\phi}}$, it is obtain by running one step PFODE (Eq. (3)) from $t_k'$ to $t_{k+1}'$ with initial distribution $Y_{t_k'}$.

Recently, Dou et al. discrete the interval $[t_k', t_{k+1}']$ in $M$ smaller interval and run multi step PFODE to obtain $\hat{Y}_{t_{k+1}'}^{\boldsymbol{\phi}}$. We note that though this operation makes theoretical analysis easier, it is far away

from the real-world application and time-consuming. Since our work aims to explain the empirical success of consistency models in application, we exactly follow the empirical operation, which does one-step PFODE instead of multi-step PFODE.

**The stepsize in the training process of consistency model.** When training the consistency model, Song et al. (2023) and Song and Dhariwal (2023) use EDM stepsize

$$t_k = (\delta + kh)^a \text{and } h = (T^{1/a} - \delta)/K \, , \tag{5}$$

with $a = 7$. As discussed in Karras et al. (2022), since VESDE has a large variance at the end of the forward process, it is more suitable for VESDE to use a large stepsize at the beginning of the reverse process instead of uniform stepsize ($a = 1$). When $a$ goes to $+\infty$, the EDM stepsize becomes theoretically friendly exponential decay stepsize $h_k = r t_k$, where $r$ is a small coefficient corresponding to accuracy parameters $\epsilon$. We note that the exponential decay stepsize is widely used in theoretical works (Chen et al., 2022a; Benton et al., 2024). In this work, we simultaneously analyze the EDM and exponential decay steps and achieve state-of-the-art discretization complexity.

**Remark 1.** *Recently, Li et al. (2024) and Dou et al. analyze the discretization complexity and estimation error bound of consistency training. As shown in Table 1, their discretization complexity is worse than our results. Furthermore, their training paradigm is different from the consistency training paradigm in application. More specifically, Li et al. (2024) use an iterative consistency training method, which trains a consistency function for each $k \in [K]$ and is time-consuming. Dou et al. only train a consistency function and embeds time $t$. However, as in the above discussion, they do multi-step PFODE in the training phase, and the empirical consistency models only do one-step PFODE. Hence, it is an interesting future work to explain the empirical success of the consistency training paradigm from the theoretical perspective under the setting in the application.*

**Notation.** We denote by $W_1$ and $W_2$ the Wasserstein distance of order one and two, respectively. Note that $W_1$ guarantee is weaker than $W_2$ guarantee since $W_1(p, q) \leq W_2(p, q)$. The push-forward operator $\sharp$ is associated with a measurable map $f : \mathcal{M}' \to \mathcal{N}$. For any measure $\mu$ over $\mathcal{M}'$, we define the push-forward measure $f \sharp \mu$ over $\mathcal{N}$ by: $f \sharp \mu(A) = \mu(f^{-1}(A))$, for any $A$ be measurable set in $\mathcal{N}$. Before introducing our theoretical guarantee, we first organize the notation.

Diffusion models:

- Let $(X_t)_{t \in [0,T]}$ be the random variable of the forward process (Eq. (1)). We define by $\delta = t_0 \leq t_1 \leq \cdots \leq t_K = T$ and $h_k = t_k - t_{k-1}$ the discretization points and stepsize.
- Let $t' = T - t$ be the reverse time and $(Y_{t'})_{t' \in [0,T]} = (X_{T-t'})_{t' \in [0,T]}$ be the random variable of reverse process (Eq. (2)). We define by $t'_k = T - t_{K-k}$ and $h'_k = h_{K-k}$ the discretization points and stepsize in the reverse process.
- Let $p_K$ be the distribution generated by running the discrete process Eq. (3) with $s_\phi$, the complexity of the sample is the requirement of $K$ to guarantee $W_2(p_K, q_0) \leq \epsilon_{W_2}$.

Consistency models:

- The goal of consistency models is to learn a consistency function $\boldsymbol{f_\theta}(Y, t')$ to directly map pure noise $Y \sim \mathcal{N}(0, T^2 I_d)$ and $t' = 0$ (the start of the reverse process) to $q_0$.
- We denote by $\boldsymbol{f}_{\boldsymbol{\theta},0} \sharp \mathcal{N}(\mathbf{0}, T^2 I_d)$ the generated distribution of the above operation. Since the consistency function is one step, the discretization complexity is the requirement of $K$ in the training process (Eq. (4)) to guarantee $W_2(\boldsymbol{f}_{\boldsymbol{\theta},0} \sharp \mathcal{N}(\mathbf{0}, T^2 I_d), q_0) \leq \epsilon_{W_2}$.

## 4 DISCRETIZATION COMPLEXITY OF CONSISTENCY MODEL IN APPLICATION

This section provides SOTA discretization complexity for the training phase of consistency models. Before showing our results, we introduce some suitable assumptions on data distribution, pre-trained score function, and consistency function.

**Assumption 1.** *$q_0$ is supported on a compact set $\mathcal{M}$ and $0 \in \mathcal{M}$.*

We define by $R$ the diameter of the compact $R = \sup\{\|x - y\|_2 : x, y \in \mathcal{M}\}$ and assume $R > 1$. The bounded support assumption is support by much empirical evidence (Pope et al., 2021; Tang

and Yang, 2024) and is satisfied by the image dataset. Currently, this assumption is widely used by current theoretical work (De Bortoli, 2022; Chen et al., 2022b; Lyu et al., 2024) and is the most lightweight assumption for VE-based method (Yang et al., 2024).

For the approximated score function, different from Lyu et al. (2024), which considers the VPSDE forward process, we consider a $L_2$ approximated score function depends on the noise $\sigma_t^2$ since VESDE forward process will have a larger $\sigma_t^2$ compared with VPSDE. As discussed in Chen et al. (2022a) (Remark 1), this assumption matches the order of the ground truth score function.

**Assumption 2.** *There exists a constant $\epsilon_{\text{score}}$ such that for any $k \in [K]$,*

$$\mathbb{E}_{X_{t_k} \sim q_{t_k}} \left[ \|s_\phi(X_{t_k}, t_k) - \nabla \log q_{t_k}(X_{t_k})\|_2^2 \right] \leq \epsilon_{\text{score}}^2 / \sigma_{t_k}^2 .$$

We also assume the consistency function is accurate enough. Since we analyze the consistency distillation paradigm, we assume after the one-step reverse PFODE process, the results of the learned consistency function are still close.

**Assumption 3.** *There exists a constant $\epsilon_{\text{cm}}$ such that for any $k \in [K]$*

$$\mathbb{E}_{Y_{t'_k} \sim q_{t'_k}} \left[ \left\| \boldsymbol{f_\theta}\left(Y_{t'_k}, t'_k\right) - \boldsymbol{f_\theta}\left(\hat{Y}_{t'_{k+1}}^\phi, t'_{k+1}\right) \right\|_2^2 \right] \leq \epsilon_{\text{cm}}^2 \left(t'_{k+1} - t'_k\right)^2 .$$

The above assumption is exactly the same as the one in Lyu et al. (2024). When considering the consistency training paradigm, Li et al. (2024) also decouple learning and generation processes and assume $\|\boldsymbol{f_\theta}(X_t, t) - \boldsymbol{f}^{\text{ex}}(X_t, t)\|_2^2$ is small enough.

Similar with previous theoretical analysis (Lyu et al., 2024; Li et al., 2024; Dou et al.), we also assume $\boldsymbol{f_\theta}(Y, t')$ is Lipschitz. Different from previous works, we assume the Lipschitz constant depends on $t$. To determine the true order of the Lipschitz constant, we first recall the result of the ground-truth $\nabla \log q_t(\cdot)$ and $\nabla^2 \log q_t(\cdot)$. Let $\mathbf{m}_t(X_t) := \mathbb{E}_{q_{0|t}(\cdot|X_t)}[X_0]$ and $\boldsymbol{\Sigma}_t(X_t) := \text{Cov}_{q_{0|t}(\cdot|\mathbf{x}_t)}(X_0)$ be the posterior mean and variance. Benton et al. (2024) achieve the following result.

**Lemma 1.** *[The VESDE version of Lemma 5.(Benton et al., 2024)] Considering VESDE forward process Eq.* (1), *for all $t \geq 0$ and $\forall X_t \in \mathbb{R}^d$, we have that*

$$\nabla \log q_t(X_t) = \frac{\mathbf{m}_t - X_t}{\sigma_t^2} \text{ and } \nabla^2 \log q_t(X_t) = -\sigma_t^{-2} I_d + \sigma_t^{-4} \boldsymbol{\Sigma}_t .$$

It is clear that $\mathbf{m}_t$ directly maps the noised data to the clean target data distribution, which is $\boldsymbol{f}^{\text{ex}}$ in our work. Then, the ground-truth score function can be parameterized as $\nabla \log q_{T-t'}(Y, T - t') = (\boldsymbol{f}^{\text{ex}}(Y, t') - Y)/\sigma_{T-t'}^2$. We note that this parametrization is widely used in applications. For example, Karras et al. (2022) and Kim et al. (2023) parameterize the approximated score $s_\phi(X_t, t) = (D_\phi(X_t, t) - X_t)/\sigma_t^2$ and using the score matching algorithm to learn $D_\phi$ (Eq. (2,3) of Karras et al. (2022)), where $D_\phi(X_t, t)$ is the denoised auto-encoder (DAE). After that, they use $s_\phi$ and $K$ discrete steps to generate samples (Line 7-9 of Algorithm 2, (Karras et al., 2022)).

Using the above parametrization and the second result of Lemma 1, we know that

$$\nabla^2 \log q_{T-t'}(Y, T - t') = \frac{\nabla \boldsymbol{f}^{\text{ex}}(Y, t') - I_d}{\sigma_{T-t'}^2} = -\sigma_{T-t'}^{-2} I_d + \sigma_{T-t'}^{-4} \boldsymbol{\Sigma}_{T-t'} .$$

Then, we know that $\nabla \boldsymbol{f}^{\text{ex}}(Y, t') = \boldsymbol{\Sigma}_{T-t'}/\sigma_{T-t'}^2$. With the bounded support assumption, we know that $\|\boldsymbol{\Sigma}_{T-t'}\|_{\text{op}} \leq R^2$ for $\forall t' \in [0, T - \delta]$, which indicates the true order of the Lipschitz constant of $\boldsymbol{f}^{\text{ex}}(Y, t')$ has order $\|\nabla \boldsymbol{f}^{\text{ex}}(Y, t')\|_{\text{op}} \leq R^2/\sigma_{T-t}^2$.

**Example 1.** To make a clearer discussion, we use a Gaussian distribution $X_0 \sim \mathcal{N}(\mu, \Sigma)$ as an example. More specifically, as shown in Guo et al. (2024) (Lemma 1), under this setting, we have that

$$\mathbb{E}[X_0|X_t] = \mu + \left(\Sigma + \sigma_t^2 I_d\right)^{-1} \Sigma (X_t - \mu) .$$

It is clear that $\nabla_{X_t}\mathbb{E}[X_0|X_t] = \left(\Sigma + \sigma_t^2 I_d\right)^{-1} \Sigma$, which has the order $1/\sigma_t^2$.

Hence, the following assumption is more realistic since it matches the true order of $\boldsymbol{f}^{\text{ex}}(Y, t')$.

**Assumption 4.** $\boldsymbol{f_\theta}(Y, t'_k)$ is $L_{f,t'_k}$ Lipschitz for $\forall k \in [K]$, where $L_{f,t} = R^2/\sigma_{T-t}^2$.

**Remark 2.** *Recently, when considering the analysis of consistency models, some works (Lyu et al., 2024; Li et al., 2024; Dou et al.) assume the second moment of data distribution $\mathbb{E}[\|q_0\|_2^2]$ is bounded, which is a slightly weaker than* **Assumption 1**. *The dependence of $R$ comes from the time-dependent Lipschitz constant of $\boldsymbol{f_\theta}$, which matches the true order. As shown in Section 4.2,* **Assumption 4** *is necessary to achieve an improved discretization complexity. We also note that to do a refined analysis for consistency models, Lemma 3.13 of Lyu et al. (2024) and Lemma D.2 of Dou et al. also assume* **Assumption 1** *to obtain the Lipschcitz constant $R^2/\sigma_{T-t}^4$ for the score function.*

With these assumptions, we obtain the following results under the VESDE and EDM stepsize, which explain the success of empirical consistency models from the discretization complexity perspective.

**Theorem 1.** *Assume* **Assumption 1***, 2, 3, 4 holds and consider the EDM stepsize (5). Then, the one-step generation error is bounded by*

$$W_2\left(\boldsymbol{f_{\theta,0}}\sharp\mathcal{N}\left(\mathbf{0},T^2I_d\right),q_0\right)\lesssim\frac{R^3}{T^2}+\frac{R^2d(T/\delta)^{\frac{1}{2a}}}{\sqrt{K}\delta}+\frac{R^2\epsilon_{\text{score}}\log(T/\delta)}{\delta}+\epsilon_{\text{cm}}T+\sqrt{d}\delta\,.$$

*where $\boldsymbol{f_{\theta,0}}$ is the learned consistency function at the reverse time $t'=0$ (the forward process $t=T$). Furthermore, by choosing $T\geq R^{1.5}/\sqrt{\epsilon_{W_2}}$, $\delta=\epsilon_{W_2}/\sqrt{d}$, $\epsilon_{\text{cm}}\leq\epsilon_{W_2}/T$ and $\epsilon_{\text{score}}\leq\epsilon_{W_2}\delta/(R^2\log(T/\delta))$, the output is $\epsilon_{W_2}$-close to $q_0$ with discretization complexity*

$$K=O\left(R^{4+\frac{3}{2a}}d^{3+\frac{1}{2a}}/\epsilon_{W_2}^{4+\frac{3}{2a}}\right)$$

Song et al. (2023) choose $a=7$ as the parameter of the EDM scheme, which leads to $O(1/\epsilon_{W_2}^{59/14})$ complexity. When considering exponential decay stepsize, we can improve the results.

**Corollary 1.** *Assume* **Assumption 1***, 2, 3, 4 holds and consider the exponential decay stepsize $h_k=rt_k$ for $\forall k\in[1,K]$, where $r=\epsilon_{W_2}^4/\left(R^4d^3\log^2(T/\delta)\right)$. Then, we have that*

$$W_2\left(\boldsymbol{f_{\theta,0}}\sharp\mathcal{N}\left(\mathbf{0},T^2I_d\right),q_0\right)\lesssim\frac{R^3}{T^2}+\frac{R^2d\log^{1.5}(T/\delta)}{\sqrt{K}\delta}+\frac{\epsilon_{\text{score}}\log(T/\delta)}{\delta}+\epsilon_{\text{cm}}T+\sqrt{d}\delta\,.$$

*By choosing $T\geq R^{1.5}/\sqrt{\epsilon_{W_2}}$, $\delta=\epsilon_{W_2}/\sqrt{d}$, $\epsilon_{\text{cm}}\leq\epsilon_{W_2}/T$ and $\epsilon_{\text{score}}\leq\epsilon_{W_2}\delta/(R^2\log(T/\delta))$, the output is $\epsilon_{W_2}$-close to $q_0$ with discretization complexity $K=O\left(R^4d^3\log^3(T/\delta)/\epsilon_{W_2}^4\right)$.*

As shown in Table 1, the above discretization complexity significantly improves the current consistency model results by taking full use of the time-dependent Lipschitz constant (Section 4.2). Compared with diffusion models, we achieve competitive and even better discretization complexity for both reverse SDE and reverse PFODE settings (Table 1).

### 4.1 Multi-step Sampling Reduce the Discretization Complexity

To achieve better performance, Song et al. (2023) also provide a multi-step sampling method. Let $T=\tau_1\geq\tau_2\geq...\geq\tau_N$ be a sequence of time points , $p_1$ be the one-step generated distribution $\boldsymbol{f_{\theta,0}}\sharp\mathcal{N}\left(\mathbf{0},T^2I_d\right)$ and $X^{\tau_1}\sim p_1$. The $n$-step sampling process follows the following procedure:

$$X^{\tau_n}=\boldsymbol{f_\theta}(X^{\tau_{n-1}}+\sigma_{\tau_n}Z,\tau_n)\,,Z\sim\mathcal{N}(0,I)\,. \tag{6}$$

which first adds noise to the $(n-1)$-step sampling data using the VESDE forward process and then generates $X^{\tau_n}$. Let $p_n$ be law $(X^{\tau_n})$ Recently, Lyu et al. (2024) make an important step in understanding the multi-step sampling mechanism in consistency models and prove that this operation can reduce the $W_2$ error with a suitable $N$. However, as shown in Corollary 3.14 of Lyu et al. (2024), the discretization complexity of one-step and multi-step sampling are both $\tilde{O}(L_f/\epsilon_{W_2}^7)$, which means the multi-step sampling can not reduce the requirement of discretization. Hence, we need a more refined analysis under a realistic setting to show the role of multi-step sampling. Since $n=2$ is enough to generate high-quality samples in application (Song et al., 2023), we analyze 2-step sampling and improve the discretization complexity with a designed $\tau_2$.

**Corollary 2.** *Assume* **Assumption 1***, 2, 3, 4 holds and consider the EDM stepsize. Then, for 2-step generation, we have that*

$$W_2\left(p_2,q_0\right)\lesssim\sqrt{d}\delta+\frac{R^5}{\tau_2^2T^2}+\frac{R^2d(T/\delta)^{\frac{1}{2a}}}{\sqrt{K}\delta}+\frac{\left(R^2\log(T/\delta)/\tau_2^2+\log(\tau_2/\delta)\right)R^2\epsilon_{\text{score}}}{\delta}+\left(\frac{R^2T}{\tau_2^2}+\tau_2\right)\epsilon_{\text{cm}}\,.$$

The error bound comes from three sources: the early stopping term, the previous error $W_2(p_1, q_\delta)$ and the discretization error at this sampling phase. The core observation is that the multi-step sampling can reduce the requirement of $T$ due to the $R^5/(\tau_2^2 T^2)$ term [3], which further improve the discretization complexity in the training phase and the dependence on $\epsilon_{\text{score}}$ and $\epsilon_{\text{cm}}$. At the end of this part, we discuss the choice of $\tau_2$, which depends on the different error terms we focus on. We note our analysis can be directly extended to multi-step sampling. Since Song et al. (2023) show that 2-step sampling is enough for consistency models (Table 1, the NFE number is the sampling number) to achieve great performance , we use 2-step sampling as an example for a clearer discussion.

**Case 1: The large learning error.** As shown in Theorem 1, to achieve the $\epsilon_{W_2}$ guarantee, $\epsilon_{\text{cm}}$ is required to be smaller than $\epsilon_{W_2}^{1.5}/R^{1.5}$. We note that after 2-step sampling, the coefficient of $\epsilon_{\text{cm}}$ becomes $R^2 T/\tau_2^2 + \tau_2$, which can be smaller than $T$. For example, with $\tau_2 = \sqrt{T}$, we require $T \geq R^{\frac{5}{3}}/\epsilon_{W_2}^{\frac{1}{3}}$ to guarantee $R^5/\tau_2^2 T^2$ smaller than $\epsilon_{W_2}$. Then, we have that

$$K = O\left(R^{4+\frac{5}{3a}} d^{3+\frac{1}{2a}}/\epsilon_{W_2}^{4+\frac{4}{3a}}\right) \text{ and } \epsilon_{\text{cm}} \leq \min\left\{\epsilon_{W_2}/R^2, \epsilon_{W_2}^{\frac{7}{6}}/R^{\frac{5}{6}}\right\}.$$

We note that the above discretization complexity is better Theorem 1 and the requirement of $\epsilon_{\text{cm}}$ is weaker than Theorem 1, which show that multi-step sampling can effectively improve the results.

**Case 2: The large discretization error.** In this case, we focus on a well-learned consistency function setting, which means $\epsilon_{\text{cm}}$ is small, and the dominant term is the discretization error. In this setting, we choose $\tau_2 = T/2$ and have that

$$K = O\left(R^{4+\frac{5}{4a}} d^{3+\frac{1}{2a}}/\epsilon_{W_2}^{4+\frac{5}{4a}}\right) \text{ and } \epsilon_{\text{cm}} \leq \epsilon_{W_2}^{\frac{4}{4}}/R^{\frac{5}{4}}.$$

It is clear that the discretization complexity is better, and the requirement of $\epsilon_{\text{cm}}$ is higher than the one in case 1. Hence, it would be better to choose a suitable starting point $\tau_2$ depending on the large error term. In application, Song et al. (2023) use a greedy policy to find $\tau_2$ with the lowest FID. It is an interesting future work to design an optimal $\tau_2$ from the theoretical perspective.

## 4.2 PROOF SKETCH AND TECHNIQUE NOVELTY

In this section, we first provide a proof sketch in the first paragraph, which is similar to Lyu et al. (2024). Then, we highlight our technique novelty to take advantage of the time-dependent Lipschitz constant to achieve the state-of-the-art discretization complexity.

**Proof Sketch.** We first decompose $W_2\left(\boldsymbol{f}_{\boldsymbol{\theta},0}\sharp\mathcal{N}\left(\mathbf{0}, T^2 I_d\right), q_0\right)$:

$$W_2\left(\boldsymbol{f}_{\boldsymbol{\theta},0}\sharp\mathcal{N}\left(\mathbf{0}, T^2 I_d\right), \boldsymbol{f}_{\boldsymbol{\theta},0}\sharp q_T\right) + W_2\left(\boldsymbol{f}_{\boldsymbol{\theta},0}\sharp q_T, q_\delta\right) + W_2\left(q_\delta, q_0\right),$$

where the first is due to the forward process, the second is the discretization error, and the third is due to the early stopping technique. In this part, we focus on the most challenging discretization term:

$$\left(\mathbb{E}_{Y_0\sim q_T}\left[\left\|\boldsymbol{f}_{\boldsymbol{\theta}}\left(Y_0, 0\right) - \boldsymbol{f}^{\text{ex}}\left(Y_0, 0\right)\right\|_2^2\right]\right)^{1/2}$$

$$= \left(\mathbb{E}_{Y_0\sim q_T}\left[\left\|\sum_{k=0}^{K-1}\left(\boldsymbol{f}_{\boldsymbol{\theta}}\left(Y_{t'_k}, t'_k\right) - \boldsymbol{f}_{\boldsymbol{\theta}}\left(Y_{t'_{k+1}}, t'_{k+1}\right)\right)\right\|_2^2\right]\right)^{1/2}$$

$$\leq \sum_{k=0}^{K-1}\left(\mathbb{E}_{Y_0\sim q_T}\left[\left\|\boldsymbol{f}_{\boldsymbol{\theta}}\left(Y_{t'_k}, t'_k\right) - \boldsymbol{f}_{\boldsymbol{\theta}}\left(\hat{Y}_{t'_{k+1}}^\phi, t'_{k+1}\right) + \boldsymbol{f}_{\boldsymbol{\theta}}\left(\hat{Y}_{t'_{k+1}}^\phi, t'_{k+1}\right) - \boldsymbol{f}_{\boldsymbol{\theta}}\left(Y_{t'_{k+1}}, t'_{k+1}\right)\right\|_2^2\right]\right)^{1/2}$$

$$\leq \sum_{k=0}^{K-1}\left(\mathbb{E}_{Y_0\sim q_T}\left[\left\|\boldsymbol{f}_{\boldsymbol{\theta}}\left(\hat{Y}_{t'_{k+1}}^\phi, t'_{k+1}\right) - \boldsymbol{f}_{\boldsymbol{\theta}}\left(Y_{t'_{k+1}}, t'_{k+1}\right)\right\|_2^2\right]\right)^{1/2} + \text{approximated } \boldsymbol{f}_{\boldsymbol{\theta}} \text{ error}$$

$$\leq \sum_{k=0}^{K-1} L_{f,t_{k+1}}\left(\mathbb{E}_{Y_0\sim q_T}\left[\left\|\hat{Y}_{t_{k+1}}^\phi - Y_{t'_{k+1}}\right\|_2^2\right]\right)^{1/2} + \text{approximated } \boldsymbol{f}_{\boldsymbol{\theta}} \text{ error},$$

---

[3]At the remaining part of this section, we focus on $\epsilon_{W_2}$ since it is the dominated term of the discretization complexity

where the first equality follows the fact that $\boldsymbol{f}^{\text{ex}}(Y_0, 0) = X_\delta = \boldsymbol{f_\theta}(Y_{T-\delta}, T-\delta)$. By using Grönwall's inequality, we can obtain the error bound of a single step PFODE:

$$\left(\mathbb{E}_{Y_0 \sim q_T}\left[\left\|\hat{Y}_{t_{k+1}}^\phi - Y_{t'_{k+1}}\right\|_2^2\right]\right)^{1/2} \lesssim \frac{dh_k'^{1.5}}{\sqrt{T - t_k'}} + h_k' \epsilon_{\text{score}}.$$

Then, the remaining term is to control $\sum_{k=0}^{K-1} L_{f,t_{k+1}} \left(\frac{dh_k'^{1.5}}{\sqrt{T-t_k'}} + h_k'\epsilon_{\text{score}}\right)$. In the following paragraphs, we discuss why the time-dependent Lipschitz is suitable for EDM stepsize and is necessary to achieve competitive results compared with the well-studied diffusion models.

**The time-dependent Lipschitz constant reduce the influence of $T$ and $\delta$.** At the beginning of this paragraph, we first show the results of uniform Lipschtiz constant.

$$dL_f \sum_{k=0}^{K-1} \frac{h_k^{1.5}}{\sqrt{t_k}} \asymp dL_f h^{0.5} \sum_{k=0}^{K-1} \frac{h_k}{t_k^{1/2a}} \asymp dL_f h^{0.5} \int_\delta^T \frac{1}{t^{\frac{1}{2a}}} \, \mathrm{d}t \le \frac{dL_f T^{1/2a}}{\sqrt{K}} T^{\frac{2a-1}{2a}} = \frac{dL_f T}{\sqrt{K}},$$

where the first equality follows the fact that $\frac{h_k}{h} \asymp t_k^{\frac{a-1}{a}}$ and the inequality comes from $h = (T^{\frac{1}{a}} - \delta)/K$. Since this constant needs to be hold for $\forall t' \in [0, T-\delta]$, we choose the uniform constant $L_{f,T-\delta} = R^2/\delta^2$ and the discretization complexity is $K = O\left(R^7 d^{5.5}/\epsilon_{W_2}^7\right)$. It is clear that a uniform Lipschitz constant prevents an improved $K$ with the EDM stepsize. However, Karras et al. (2022) has shown that a suitable $a = 7$ can significantly improve the generation quality and consistency models adopt this setting. In the following part, we show that with a time-dependent $L_{f,t}$, the discretization can be improved by using EDM stepsize. With $L_{f,t} = R^2/\sigma_{T-t}^2$, we have

$$\frac{dR^2}{\delta} \sum_{k=1}^{K} \frac{h_k^{1.5}}{t_k^{1.5}} \asymp \frac{dR^2 h^{0.5}}{\delta} \sum_{k=1}^{K} \frac{h_k}{t_k^{\frac{2a+1}{2a}}} \asymp \frac{dR^2 h^{0.5}}{\delta} \asymp \frac{dR^2 h^{0.5} \delta^{-\frac{1}{2a}}}{\delta} \asymp \frac{dR^2 (T/\delta)^{\frac{1}{2a}}}{\delta\sqrt{K}}.$$

Then, we obtain the discretization complexity $K = O\left(R^{4+\frac{3}{2a}} d^{3+\frac{1}{2a}}/\epsilon_{W_2}^{4+\frac{3}{2a}}\right)$, which is better than the results with uniform Lipschitz constant and show the influence of EDM parameter $a$.

## 5 CONCLUSION

In this work, we make the first step to explain why consistency models perform well from a theoretical perspective. More specifically, we bridge the gap between theory and application by analyzing the discretization complexity of the consistency model with the VESDE forward process and EDM discretization scheme. Under this realistic setting with great empirical performance, we first achieve the state-of-the-art discretization complexity $O\left(1/\epsilon_{W_2}^{4+\frac{3}{2a}}\right)$ by making full use of time-dependent Lipschitz constant. Then, we analyze the improved multi-step sampling algorithm proposed by Song et al. (2023) and show that this algorithm improves discretization complexity $O\left(1/\epsilon_{W_2}^{4+\frac{5}{4a}}\right)$. Finally, to explore the theoretical boundary of consistency models, we further analyze consistency models with exponential decay stepsize and achieve $O\left(1/\epsilon_{W_2}^4\right)$ complexity.

In conclusion, we achieve competitive results for consistency models compared with diffusion models under the realistic setting, which shows the potential of consistency models.

**Future work and limitation.** In this work, we directly assume the approximated score and consistency function are accurate enough. For the approximated score function, some works analyze its learning process (Chen et al., 2023b; Yuan et al., 2023). For the consistency function, Dou et al. analyze its estimation error. However, as discussed in Section 3.2, they run multi-step PFODE instead of one-step PFODE, which does not match the empirical operation and is time-consuming. Hence, it is interesting to analyze the learning process of consistency models under a realistic setting and achieve an end-to-end analysis.

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

APPENDIX

## A  THE ANALYSIS FOR CONSISTENCY MODEL

**Theorem 1.** *Assume* **Assumption 1**, *2, 3, 4 holds and consider the EDM stepsize (5). Then, the one-step generation error is bounded by*

$$W_2\left(\boldsymbol{f}_{\boldsymbol{\theta},0}\sharp\mathcal{N}\left(\mathbf{0},T^2I_d\right),q_0\right) \lesssim \frac{R^3}{T^2} + \frac{R^2 d(T/\delta)^{\frac{1}{2a}}}{\sqrt{K}\delta} + \frac{R^2\epsilon_{\text{score}}\log(T/\delta)}{\delta} + \epsilon_{\text{cm}}T + \sqrt{d}\delta\,.$$

*where $\boldsymbol{f}_{\boldsymbol{\theta},0}$ is the learned consistency function at the reverse time $t' = 0$ (the forward process $t = T$). Furthermore, by choosing $T \geq R^{1.5}/\sqrt{\epsilon_{W_2}}$, $\delta = \epsilon_{W_2}/\sqrt{d}$, $\epsilon_{\text{cm}} \leq \epsilon_{W_2}/T$ and $\epsilon_{\text{score}} \leq \epsilon_{W_2}\delta/(R^2\log(T/\delta))$, the output is $\epsilon_{W_2}$-close to $q_0$ with discretization complexity*

$$K = O\left(R^{4+\frac{3}{2a}}d^{3+\frac{1}{2a}}/\epsilon_{W_2}^{4+\frac{3}{2a}}\right)$$

**Proof.** We first decompose $W_2\left(\boldsymbol{f}_{\boldsymbol{\theta},0}\sharp\mathcal{N}\left(\mathbf{0},T^2I_d\right),q_0\right)$:

$$W_2\left(\boldsymbol{f}_{\boldsymbol{\theta},0}\sharp\mathcal{N}\left(\mathbf{0},T^2I_d\right),q_0\right) \leq W_2\left(\boldsymbol{f}_{\boldsymbol{\theta},0}\sharp\mathcal{N}\left(\mathbf{0},T^2I_d\right),\boldsymbol{f}_{\boldsymbol{\theta},0}\sharp q_T\right) + W_2\left(\boldsymbol{f}_{\boldsymbol{\theta},0}\sharp q_T,q_\delta\right) + W_2\left(q_\delta,q_0\right)\,,$$

where the first term is the reverse beginning error due to the forward process, the second term is the discretization error due to the discretization training, and the third term is due to the early stopping technique. We first define a joint distribution $\gamma \in \Gamma\left(\mathcal{N}\left(\mathbf{0},T^2I_d\right),q_T\right)$ between $\mathcal{N}\left(\mathbf{0},T^2I_d\right)$ and $q_T$ and take a couple of $(\bar{Y}_0,Y_0) \sim \gamma$, which indicates

$$\int_{\mathbb{R}^d}\gamma(\cdot,Y_0)\mathrm{d}Y_0 = \mathcal{N}\left(\mathbf{0},T^2I_d\right)$$
$$\int_{\mathbb{R}^d}\gamma(\bar{Y}_0,\cdot)\mathrm{d}\bar{Y}_0 = q_T\,.$$

Then, we have that

$$W_2\left(\boldsymbol{f}_{\boldsymbol{\theta},0}\sharp\mathcal{N}\left(\mathbf{0},T^2I_d\right),q_\delta\right)$$
$$\leq \left(\mathbb{E}_\gamma\left[\left\|\boldsymbol{f}_{\boldsymbol{\theta}}\left(\bar{Y}_0,0\right) - \boldsymbol{f}^{\text{ex}}\left(Y_0,0\right)\right\|_2^2\right]\right)^{1/2}$$
$$\leq \left(\mathbb{E}_\gamma\left[\left\|\boldsymbol{f}_{\boldsymbol{\theta}}\left(\bar{Y}_0,0\right) - \boldsymbol{f}_{\boldsymbol{\theta}}\left(Y_0,0\right)\right\|_2^2\right]\right)^{1/2} + \left(\mathbb{E}_\gamma\left[\left\|\boldsymbol{f}_{\boldsymbol{\theta}}\left(Y_0,0\right) - \boldsymbol{f}^{\text{ex}}\left(Y_0,0\right)\right\|_2^2\right]\right)^{1/2}$$
$$\leq L_{f,0}\left(\mathbb{E}_\gamma\left[\left\|\bar{Y}_0 - Y_0\right\|_2^2\right]\right)^{1/2} + \left(\mathbb{E}_\gamma\left[\left\|\boldsymbol{f}_{\boldsymbol{\theta}}\left(Y_0,0\right) - \boldsymbol{f}^{\text{ex}}\left(Y_0,0\right)\right\|_2^2\right]\right)^{1/2}$$
$$\leq \frac{R^2}{T^2}W_2(\mathcal{N}(0,T^2I_d),q_T) + \left(\mathbb{E}_\gamma\left[\left\|\boldsymbol{f}_{\boldsymbol{\theta}}\left(Y_0,0\right) - \boldsymbol{f}^{\text{ex}}\left(Y_0,0\right)\right\|_2^2\right]\right)^{1/2}$$
$$= \frac{R^2}{T^2}W_2(\mathcal{N}(0,T^2I_d),q_T) + \left(\mathbb{E}_{Y_0\sim q_T}\left[\left\|\boldsymbol{f}_{\boldsymbol{\theta}}\left(Y_0,0\right) - \boldsymbol{f}^{\text{ex}}\left(Y_0,0\right)\right\|_2^2\right]\right)\,.$$

where the first inequality follow the fact $f^{\text{ex}}\left(Y_0,0\right) = X_\delta$ when $Y_0 \sim q_T$ and the last inequality follows the fact that $\gamma$ can be any coupling between $\mathcal{N}(0,T^2I_d)$ and $q_T$.

**The reverse beginning error.**  We first control the reverse beginning error term. When considering the $W_2$ guarantee, we have that

$$W_2(\mathcal{N}(0,T^2I_d),q_T) \leq \left(\mathbb{E}_{q_0}\left[\left\|X_0 + \left(T - T\right)\xi\right\|_2^2\right]\right)^{1/2} \leq R\,.$$

**The discretization error.** In this section, we control the discretization error:

$$\left(\mathbb{E}_{Y_0 \sim q_T}\left[\left\|\boldsymbol{f_\theta}\left(Y_0, 0\right) - \boldsymbol{f}^{\mathrm{ex}}\left(Y_0, 0\right)\right\|_2^2\right]\right)^{1/2}$$

$$= \left(\mathbb{E}_{Y_0 \sim q_T}\left[\left\|\sum_{k=0}^{K-1}\left(\boldsymbol{f_\theta}\left(Y_{t'_k}, t'_k\right) - \boldsymbol{f_\theta}\left(Y_{t'_{k+1}}, t'_{k+1}\right)\right)\right\|_2^2\right]\right)^{1/2}$$

$$\leq \sum_{k=0}^{K-1}\left(\mathbb{E}_{Y_0 \sim q_T}\left[\left\|\boldsymbol{f_\theta}\left(Y_{t'_k}, t'_k\right) - \boldsymbol{f_\theta}\left(Y_{t'_{k+1}}, t'_{k+1}\right)\right\|_2^2\right]\right)^{1/2}$$

$$= \sum_{k=0}^{K-1}\left(\mathbb{E}_{Y_0 \sim q_T}\left[\left\|\boldsymbol{f_\theta}\left(Y_{t'_k}, t'_k\right) - \boldsymbol{f_\theta}\left(\hat{Y}^\phi_{t'_{k+1}}, t'_{k+1}\right) + \boldsymbol{f_\theta}\left(\hat{Y}^\phi_{t'_{k+1}}, t'_{k+1}\right) - \boldsymbol{f_\theta}\left(Y_{t'_{k+1}}, t'_{k+1}\right)\right\|_2^2\right]\right)^{1/2}$$

$$\leq \sum_{k=0}^{K-1}\left(\mathbb{E}_{Y_0 \sim q_T}\left[\left\|\boldsymbol{f_\theta}\left(Y_{t'_k}, t'_k\right) - \boldsymbol{f_\theta}\left(\hat{Y}^\phi_{t'_{k+1}}, t'_{k+1}\right)\right\|_2^2\right]\right)^{1/2}$$

$$+ \sum_{k=0}^{K-1}\left(\mathbb{E}_{Y_0 \sim q_T}\left[\left\|\boldsymbol{f_\theta}\left(\hat{Y}^\phi_{t'_{k+1}}, t'_{k+1}\right) - \boldsymbol{f_\theta}\left(Y_{t'_{k+1}}, t'_{k+1}\right)\right\|_2^2\right]\right)^{1/2} := E_1 + E_2\,,$$

where the first inequality follows the fact that $\boldsymbol{f}^{\mathrm{ex}}\left(Y_0, 0\right) = X_\delta = \boldsymbol{f_\theta}\left(Y_{T-\delta}, T - \delta\right)$. For term $E_1$, since we assume the learned consistency model is accurate enough (**Assumption 3**), then we have that:

$$E_1 = \sum_{k=0}^{K-1}\left(\mathbb{E}_{Y_0 \sim q_T}\left[\left\|\boldsymbol{f_\theta}\left(Y_{t'_k}, t'_k\right) - \boldsymbol{f_\theta}\left(\hat{Y}^\phi_{t'_{k+1}}, t'_{k+1}\right)\right\|_2^2\right]\right)^{1/2}$$

$$= \sum_{k=0}^{K-1}\left(\mathbb{E}_{Y_{t'_k} \sim q_{t'_k}}\left[\left\|\boldsymbol{f_\theta}\left(Y_{t'_k}, t'_k\right) - \boldsymbol{f_\theta}\left(\hat{Y}^\phi_{t'_{k+1}}, t'_{k+1}\right)\right\|_2^2\right]\right)^{1/2}$$

$$\leq \epsilon_{\mathrm{cm}}\sum_{k=0}^{K-1} h'_k = \epsilon_{\mathrm{cm}}\left(T - \delta\right)\,.$$

For term $E_2$, we know that

$$E_2 = \sum_{k=0}^{K-1}\left(\mathbb{E}_{Y_0 \sim q_T}\left[\left\|\boldsymbol{f_\theta}\left(\hat{Y}^\phi_{t'_{k+1}}, t'_{k+1}\right) - \boldsymbol{f_\theta}\left(Y_{t'_{k+1}}, t'_{k+1}\right)\right\|_2^2\right]\right)^{1/2}$$

$$\leq \sum_{k=0}^{K-1} L_{f,t_{k+1}}\left(\mathbb{E}_{Y_0 \sim q_T}\left[\left\|\hat{Y}^\phi_{t'_{k+1}} - Y_{t'_{k+1}}\right\|_2^2\right]\right)^{1/2}$$

$$= \sum_{k=0}^{K-1}\frac{R^2}{\sigma_{t_{k+1}}^2}\left(\mathbb{E}_{Y_{t'_k} \sim q_{t'_k}}\left[\left\|\hat{Y}^\phi_{t'_{k+1}} - Y_{t'_{k+1}}\right\|_2^2\right]\right)^{1/2}$$

$$\lesssim \sum_{k=0}^{K-1}\frac{R^2}{(T - t'_k)^2}\left(\frac{dh'^{1.5}_k}{\sqrt{T - t'_k}} + h'_k \epsilon_{\mathrm{score}}\right)\,,$$

where the last inequality comes from Lemma 2. When considering EDM stepsize

$$t_k = (\delta + kh)^a, h = \frac{T^{\frac{1}{a}} - \delta}{K}\,,$$

we know that $\frac{h_k}{h} \asymp t_k^{\frac{a-1}{a}}$,

$$\sum_{k=1}^{K}\frac{h_k}{t_k} \asymp \int_\delta^T \frac{1}{t}\,\mathrm{d}t \asymp \log(T/\delta)\,,$$

and

$$\sum_{k=1}^{K} \frac{h_k^{1.5}}{t_k^{1.5}} \asymp h^{0.5} \sum_{k=1}^{K} \frac{h_k}{t_k^{\frac{2a+1}{2a}}} \asymp h^{0.5} \int_\delta^T \frac{1}{t^{\frac{2a+1}{2a}}} \, \mathrm{d}t \asymp h^{0.5}\delta^{-\frac{1}{2a}} \asymp \frac{(T/\delta)^{\frac{1}{2a}}}{\sqrt{K}} \,.$$

Then, term $E_2$ is control by the following inequality:

$$E_2 \le \sum_{k=1}^{K} \left( \frac{R^2 d h_k^{1.5}}{t_k^{1.5}\delta} + \frac{R^2 h_k \epsilon_{\text{score}}}{t_k \delta} \right)$$

$$\le \frac{R^2 d (T/\delta)^{\frac{1}{2a}}}{\sqrt{K}\delta} + \frac{R^2 \epsilon_{\text{score}} \log(T/\delta)}{\delta}$$

Combined with the reverse beginning, discretization and early stopping term, we have that

$$W_2 \left( \boldsymbol{f}_{\boldsymbol{\theta},0}\sharp\mathcal{N}\left(\mathbf{0}, T^2 I_d\right), q_0 \right) \lesssim \frac{R^3}{T^2} + \frac{R^2 d (T/\delta)^{\frac{1}{2a}}}{\sqrt{K}\delta} + \frac{R^2 \epsilon_{\text{score}} \log(T/\delta)}{\delta} + \epsilon_{\text{cm}} T + \sqrt{d}\delta \,.$$

To make the above inequality smaller than $\epsilon_{W_2}$, we choose $T \ge R^{1.5}/\sqrt{\epsilon_{W_2}}$, $\delta = \epsilon_{W_2}/\sqrt{d}$, $\epsilon_{\text{cm}} \le \epsilon_{W_2}/T$, and guarantee

$$\frac{R^2 d T^{\frac{1}{2a}}}{\sqrt{K}\delta^{1+\frac{1}{2a}}} \le \epsilon_{W_2} \,,$$

which indicates that $K \ge \frac{R^{4+\frac{3}{2a}} d^{2+\frac{1}{2a}}}{\epsilon_{W_2}^{4+\frac{3}{2a}}}$. After determining the discretization complexity $K$, we can also obtain the requirement of the approximated score function $\epsilon_{\text{score}} \le \epsilon_{W_2}\delta/(R^2 \log(T/\delta))$. ∎

**Corollary 1.** *Assume* **Assumption 1**, *2, 3, 4 holds and consider the exponential decay stepsize* $h_k = r t_k$ *for* $\forall k \in [1, K]$, *where* $r = \epsilon_{W_2}^4/\left(R^4 d^3 \log^2(T/\delta)\right)$. *Then, we have that*

$$W_2 \left( \boldsymbol{f}_{\boldsymbol{\theta},0}\sharp\mathcal{N}\left(\mathbf{0}, T^2 I_d\right), q_0 \right) \lesssim \frac{R^3}{T^2} + \frac{R^2 d \log^{1.5}(T/\delta)}{\sqrt{K}\delta} + \frac{\epsilon_{\text{score}} \log(T/\delta)}{\delta} + \epsilon_{\text{cm}} T + \sqrt{d}\delta \,.$$

*By choosing* $T \ge R^{1.5}/\sqrt{\epsilon_{W_2}}$, $\delta = \epsilon_{W_2}/\sqrt{d}$, $\epsilon_{\text{cm}} \le \epsilon_{W_2}/T$ *and* $\epsilon_{\text{score}} \le \epsilon_{W_2}\delta/(R^2 \log(T/\delta))$, *the output is* $\epsilon_{W_2}$-*close to* $q_0$ *with discretization complexity* $K = O\left(R^4 d^3 \log^3(T/\delta)/\epsilon_{W_2}^4\right)$.

**Proof**. For the theoretical-friendly exponential decay stepsize, we have that

$$E_2 \le \sum_{k=1}^{K=\frac{1}{r}\log(T/\delta)} \left( \frac{R^2 d h_k^{1.5}}{t_k^{2.5}} + \frac{R^2 h_k \epsilon_{\text{score}}}{t_k^2} \right)$$

$$\le \sum_{k=1}^{K=\frac{1}{r}\log(T/\delta)} \left( \frac{R^2 d h_k^{1.5}}{t_k^{1.5}\delta} + \frac{R^2 h_k \epsilon_{\text{score}}}{t_k \delta} \right) \lesssim \frac{R^2 d \log^{1.5}(T/\delta)}{\sqrt{K}\delta} + \frac{R^2 \epsilon_{\text{score}} \log(T/\delta)}{\delta} \,.$$

Combined with the reverse beginning term and discretization term, we know that

$$W_2 \left( \boldsymbol{f}_{\boldsymbol{\theta},0}\sharp\mathcal{N}\left(\mathbf{0}, T^2 I_d\right), q_0 \right) \lesssim \frac{R^3}{T^2} + \frac{R^2 d \log^{1.5}(T/\delta)}{\sqrt{K}\delta} + \frac{\epsilon_{\text{score}} \log(T/\delta)}{\delta} + \epsilon_{\text{cm}} T + \sqrt{d}\delta \,.$$

In order to guarantee the right hand of above inequality smaller than $\epsilon_{W_2}$, we choose $T \ge R^{1.5}/\sqrt{\epsilon_{W_2}}$, $\delta = \epsilon_{W_2}/\sqrt{d}$, $\epsilon_{\text{cm}}^2 \le \epsilon_{W_2}^2/T^2$, and guarantee

$$\frac{R^2 d \log^{1.5}(T/\delta)}{\sqrt{K}\delta} \le \epsilon_{W_2} \,,$$

which indicates the discretizaiton complexity is $K \ge \frac{R^4 d^3 \log^3(T/\delta)}{\epsilon_{W_2}^4} = \frac{R^4 d^3 \log^3\left(\frac{R^{1.5}\sqrt{d}}{\epsilon_{W_2}^{1.5}}\right)}{\epsilon_{W_2}^4}$. After obtaining the requirement of $K$, we can also obtain the requirement of approximated score function $\epsilon_{\text{score}} \le \epsilon_{W_2}/\log(T/\delta)$. ∎

At the end of this part, we provide the proof of multi-step sampling.

**Corollary 2.** *Assume* **Assumption 1***, 2, 3, 4 holds and consider the EDM stepsize. Then, for* 2*-step generation, we have that*

$$
W_2\left(p_2, q_0\right) \lesssim \sqrt{d}\delta + \frac{R^5}{\tau_2^2 T^2} + \frac{R^2 d(T/\delta)^{\frac{1}{2a}}}{\sqrt{K}\delta} + \frac{\left(R^2 \log(T/\delta)/\tau_2^2 + \log(\tau_2/\delta)\right) R^2 \epsilon_{\mathrm{score}}}{\delta} + \left(\frac{R^2 T}{\tau_2^2} + \tau_2\right)\epsilon_{\mathrm{cm}}.
$$

**Proof.** Take a couple of $(\boldsymbol{Y}, \boldsymbol{Z}) \sim \gamma(\boldsymbol{y}, \boldsymbol{z})$ where $\gamma \in \Gamma\left(p_{n-1}, q_\delta\right)$, take $\boldsymbol{\xi} \sim \mathcal{N}\left(\mathbf{0}, \boldsymbol{I}_d\right)$, then we have

$$
\hat{\boldsymbol{Y}} = \boldsymbol{Y} + \tau_n \boldsymbol{\xi} \sim \mu_n,
$$

$$
\hat{\boldsymbol{Z}} = \boldsymbol{Z} + \tau_n \boldsymbol{\xi} \sim q_{\tau_n}.
$$

Similar with Corollary 10 of Lyu et al. (2024), we have that

$$
\begin{aligned}
W_2\left(p_2, q_0\right) &\lesssim W_2(q_\delta, q_0) + L_{f, T-\tau_2} W_2\left(\mu_n, q_{\tau_2}\right) + \frac{R^2 d(\tau_2/\delta)^{\frac{1}{2a}}}{\sqrt{K}\delta} + \frac{R^2 \epsilon_{\mathrm{score}} \log(\tau_2/\delta)}{\delta} + \epsilon_{\mathrm{cm}}\tau_2 \\
&\lesssim W_2(q_\delta, q_0) + L_{f, T-\tau_2}\left(\mathbb{E}_\gamma \|\hat{\boldsymbol{Y}} - \hat{\boldsymbol{Z}}\|_2^2\right)^{1/2} + \frac{R^2 d(\tau_2/\delta)^{\frac{1}{2a}}}{\sqrt{K}\delta} + \frac{R^2 \epsilon_{\mathrm{score}} \log(\tau_2/\delta)}{\delta} + \epsilon_{\mathrm{cm}}\tau_2 \\
&\lesssim W_2(q_\delta, q_0) + L_{f, T-\tau_2}\left(\mathbb{E}_\gamma \|\boldsymbol{Y} - \boldsymbol{Z}\|_2^2\right)^{1/2} + \frac{R^2 d(\tau_2/\delta)^{\frac{1}{2a}}}{\sqrt{K}\delta} + \frac{R^2 \epsilon_{\mathrm{score}} \log(\tau_2/\delta)}{\delta} + \epsilon_{\mathrm{cm}}\tau_2 \\
&\lesssim \sqrt{d}\delta + \frac{R^2}{\tau_2^2}\left(\frac{R^3}{T^2} + \frac{R^2 d(T/\delta)^{\frac{1}{2a}}}{\sqrt{K}\delta} + \frac{R^2 \epsilon_{\mathrm{score}} \log(T/\delta)}{\delta} + \epsilon_{\mathrm{cm}}T\right) \\
&\quad + \frac{R^2 d(T/\delta)^{\frac{1}{2a}}}{\sqrt{K}\delta} + \frac{R^2 \epsilon_{\mathrm{score}} \log(\tau_2/\delta)}{\delta} + \epsilon_{\mathrm{cm}}\tau_2
\end{aligned}
$$

where the first line of the last inequality is introduced by the previous sampling process, and the remaining term is the discretization error of this phase.

∎

## B THE ERROR OF ONE STEP PFODE FOR THE VESDE FORWARD PROCESS

When considering the consistency distillation training paradigm, we need to run a one-step reverse PFODE starting from $Y_{t'_k}$ to obtain $\hat{Y}^\phi_{t'_{k+1}}$. Hence, we need to control one step starting from the same distribution $q$.

**Lemma 2.** *Suppose* **Assumption 1** *and* **Assumption 2** *hold and assuming* $\frac{\sigma_s^2 - \sigma_t^2}{\sigma_t^2} \leq \frac{1}{2d}$ *for any* $0 \leq t \leq s \leq T$, *then for the small interval* $t \in [t'_k, t'_{k+1}]$ *for* $\forall k \in [0, K-1]$, *we have that*

$$
W_2^2\left(qQ_{\mathrm{ODE}}^{t'_k, h'_k}, q\bar{P}_{\mathrm{ODE}}^{t'_k, h'_k}\right) \lesssim \frac{d^2 h_k'^3}{T - t'_k} + h_k'^2 \epsilon_{\mathrm{score}}^2.
$$

**Proof.** For $t \in [t'_k, t'_{k+1}]$, the reverse PFODE is

$$
\dot{Y}_t = \frac{g(T-t)^2}{2}\nabla \ln q_{T-t}\left(Y_t\right),
$$

$$
\dot{\bar{Y}}_t = \frac{g(T-t)^2}{2}s_\phi\left(\bar{Y}_{t'_k}, T - t'_k\right),
$$

for $t'_k \leq t \leq t'_{k+1}$, with $Y_{t'_k} = \bar{Y}_{t'_k} \sim q$, $Y_{t'_k + h'_k} \sim qQ_{\mathrm{ODE}}$, and $\bar{Y}_{t'_k + h'_k} \sim q\bar{P}_{\mathrm{ODE}}$. Then, we have that

$$
\begin{aligned}
\partial_t \left\|Y_t - \bar{Y}_t\right\|^2 &= 2\left\langle Y_t - \bar{Y}_t, \dot{Y}_t - \dot{\bar{Y}}_t\right\rangle \\
&= 2\left\langle Y_t - \bar{Y}_t, \frac{g(T-t)^2}{2}\left(\nabla \ln q_{T-t}\left(Y_t\right) - s_\phi\left(\bar{Y}_{t'_k}, T - t'_k\right)\right)\right\rangle \\
&\leq \frac{1}{h'_k}\left\|Y_t - \bar{Y}_t\right\|^2 + \frac{h'_k g(T-t)^4}{4}\left\|\nabla \ln q_{T-t}\left(Y_t\right) - s_\phi\left(\bar{Y}_{t'_k}, T - t'_k\right)\right\|^2
\end{aligned}
$$

As the next step, we use the Grönwall's inequality to control the one-step discretization error.

**Grönwall's inequality.** Let $\eta(\cdot)$ be a nonnegative, absolutely continuous function on $[0, h_k']$, which satisfies for a.e. $t$ the differential inequality

$$\eta'(t) \le \phi(t)\eta(t) + \psi(t)$$

where $\phi(t)$ and $\psi(t)$ are nonnegative, summable function on $[0, h_k']$. Then

$$\eta(t) \le e^{\int_0^t \phi(s)\mathrm{d}s}\left[\eta(0) + \int_0^t \psi(s)\mathrm{d}s\right].$$

By setting $\eta(t) = \left\|Y_t - \bar{Y}_t\right\|^2$ and $\psi(t) = \frac{h_k' g(T-t)^4}{4}\left\|\nabla \ln q_{T-t}(Y_t) - s_\phi\left(\bar{Y}_{t_k'}, T - t_k'\right)\right\|^2$. We note that $Y_{t_k'} = \bar{Y}_{t_k'} \sim q$ starts from the same distribution, $\eta(0) = 0$. Then, we can obtain that

$$\mathbb{E}\left[\left\|Y_{t_k'+h_k'} - \bar{Y}_{t_k'+h_k'}\right\|^2\right]$$

$$\le \exp\left(\int_0^{h_k'} \frac{1}{h_k'}\mathrm{d}t\right)\int_{t_k'}^{t_k'+h_k'} \frac{g(T-t)^4 h_k'}{4}\mathbb{E}\left[\left\|\nabla \ln q_{T-t}(Y_t) - s_\phi\left(\bar{Y}_{t_k'}, T - t_k'\right)\right\|^2\right]\mathrm{d}t$$

$$\le \int_{t_k'}^{t_k'+h_k'} g(T-t)^4 h_k'\mathbb{E}\left[\left\|\nabla \ln q_{T-t}(Y_t) - s_\phi\left(\bar{Y}_{t_k'}, T - t_k'\right)\right\|^2\right]\mathrm{d}t,$$

Then, by using 6, we have that

$$\mathbb{E}\left[\left\|Y_{t_k'+h_k'} - \bar{Y}_{t_k'+h_k'}\right\|^2\right]$$

$$\le \int_{t_k'}^{t_k'+h_k'} g(T-t)^4 h_k'\mathbb{E}\left[\left\|\nabla \ln q_{T-t}(Y_t) - s_\phi\left(\bar{Y}_{t_k'}, T - t_k'\right)\right\|^2\right]\mathrm{d}t$$

$$\lesssim \int_{t_k'}^{t_k'+h_k'} \frac{d^2 g(T-t)^4 h_k'(\sigma_{T-t_k'}^2 - \sigma_t^2)}{\sigma_{T-t}^4} + \frac{h_k' g(T-t)^4 \epsilon_{\text{score}}^2}{\sigma_{T-t}^2}\mathrm{d}t.$$

Finally, we have that

$$\mathbb{E}\left[\left\|Y_{t_k'+h_k'} - \bar{Y}_{t_k'+h_k'}\right\|^2\right] \lesssim \frac{d^2 h_k'^3}{T - t_k'} + h_k'^2 \epsilon_{\text{score}}^2.$$

∎

## C  THE DISCUSSION ON THE PREVIOUS WORK DETAIL CALCULATION OF PREVIOUS CONSISTENCY MODELS RESULTS

### C.1  THE DETAIL CALCULATION OF PREVIOUS CONSISTENCY MODELS RESULTS

In this part, we show how to replace $L_f$ with our **Assumption 4** to obtain (*) in Table 1 and discuss the reason why the noise schedule of Li et al. (2024) relies heavily on the VPSDE forward process.

**The results of Lyu et al. (2024).**   As shown in Corollary 8 of Lyu et al. (2024), the discretization complexity of consistency distillation is

$$\widetilde{O}\left(\frac{d^{1/2}R^3\left(R^6 \vee d^3\right)L_f}{\epsilon_{W_2}^7}\right).$$

Since Lyu et al. (2024) assume the Lipschitz constant of approximated consistency function holds for $t \in [\delta, T]$, we use the largest Lipschitz constant $L_{f,\max} = L_{f,T-\delta} = R^2/\sigma_\delta^2$ (**Assumption 4**).

Since Lyu et al. (2024) use VPSDE as the forward process, $\sigma_\delta^2 = \delta$ for small enough and we need $\delta = \epsilon_{W_2}^2/(\sqrt{d}(R \vee \sqrt{d}))$ to guarantee $W_2^2(q_\delta, q_0) \leq \epsilon_{W_2}^2$ (Lemma 7). Then, we obtain the discretization complexity of Lyu et al. (2024) is

$$\widetilde{O}\left(\frac{dR^5\left(R^6 \vee d^3\right)\left(R \vee \sqrt{d}\right)}{\epsilon_{W_2}^9}\right).$$

**The results of Li et al. (2024).** In this part, we first show the noise schedule of Li et al. (2024) and discuss the reason why this schedule depends heavily on VPSDE. Li et al. (2024) describe the VPSDE in a discrete perspective instead of a continuous forward process:

$$X_0 \sim q_0,$$
$$X_k = \sqrt{1-\beta_k}X_{k-1} + \sqrt{\beta_t}B_k, \quad 1 \leqslant k \leqslant K.$$

Let

$$\alpha_k := 1 - \beta_k, \quad \bar{\alpha}_k := \prod_{k'=1}^{k} \alpha_{k'}, \quad 1 \leqslant k \leqslant K.$$

Then, we know that $X_k = \sqrt{\bar{\alpha}_k}X_0 + \sqrt{1-\bar{\alpha}_k}\bar{W}_k$ for some $\bar{W}_k \sim \mathcal{N}(0, I_d)$, which indicates $X_K$ is approximately $\mathcal{N}(0, I_d)$ (VPSDE) with suitable noise schedule. Li et al. (2024) choose a specific noise schedule

$$\beta_1 = 1 - \alpha_1 = \frac{1}{K^{c_0}},$$
$$\beta_k = 1 - \alpha_k = \frac{c_1 \log K}{K}\min\left\{\beta_1\left(1 + \frac{c_1 \log K}{K}\right)^k, 1\right\}, \quad 2 \leqslant k \leqslant K,$$

where $c_0, c_1 > 0$ are large enough numerical constants. We note that when $K$ goes to $+\infty$, $\beta_k$ will goes to 0. It is quietly different from VESDE forward process since $\sigma_t^2 = t^2$ would goes to $+\infty$ when $T$ goes to $+\infty$ ($K \to +\infty$). Hence, this noise schedule heavily depends on the form of VPSDE.

After that, we show how to transfer the results of Li et al. (2024) to the results under our setting. Li et al. (2024) show the discretization complexity of consistency training paradigm with VPSDE forward process:

$$K = \widetilde{O}\left(\frac{L_f^3 d^{5/2}}{\epsilon_{W_1}}\right).$$

Similar to the above paragraph, we replace $L_f$ in Li et al. (2024) with $L_{f,\max} = \frac{R^2}{\delta}$ and $\delta = \epsilon_{W_2}^2/(\sqrt{d}(R \vee \sqrt{d}))$. Then, we require at least

$$\frac{R^6 d^{2.5}(\sqrt{d}(R \vee \sqrt{d}))^3}{\epsilon_{W_1}\epsilon_{W_2}^6}$$

discretization steps.

**The results of Dou et al..** As shown in Theorem 4.1 of Dou et al., when considering $W_1$ distance, the discretization error of ODE is $\frac{L_f d L_{\text{score}}}{\sqrt{M}}$, where $M$ is the step number discussed in Section 3.2. To make the above term smaller than $\epsilon_{W_1}$, we require

$$M \geq \frac{L_f^2 L_{\text{score}}^2 d^2}{\epsilon_{W_1}^2}.$$

Following the above discussion, We choose $L_f = L_{f,\max} = R^2/\delta$, $L_{\text{score}} = R^2/\sigma_\delta^4 = R^2/\delta^2$ and $\delta = \epsilon_{W_2}^2/(\sqrt{d}(R \vee \sqrt{d}))$. Then, the discretization complexity is

$$M \geq \frac{R^8 d^2(\sqrt{d}(R \vee \sqrt{d}))^6}{\epsilon_{W_1}^2 \epsilon_{W_2}^{12}}.$$

## C.2   THE LIPSCHITZ CONSTANT OF THE NONPARAMETRIC EMPIRICAL SCORE FUNCTION

At the beginning of this part, we first recall the time-dependent Lipschitz constant of empirical score, which is proposed by Dou et al..

**Lemma 3.** *For the mixture of Gaussian distribution $\frac{1}{n}\sum_{j=1}^{n}\mathcal{N}\left(X^j, \sigma^2 I_d\right)$, we denote $\widehat{q}$ as its density. Assume $\left\|X^j\right\|_2 \leqslant R$ for $\forall j \in [n]$, then its score function $\nabla \log \widehat{q}(\cdot)$ is L-Lipschitz continuous. Here $L = \max\left(R^2/\sigma^4, 1/\sigma^2\right)$.*

When considering VESDE forward process, the mixture of Gaussian at the forward time $t$ becomes $\frac{1}{n}\sum_{j=1}^{n}\mathcal{N}\left(X^j, \sigma_t^2 I_d\right)$. Hence, the Lipschitz constant of $\nabla \log \widehat{q}_t(\cdot)$ is $\max(R^2/\sigma_t^4, 1/\sigma_t^2)$. Since the dominated time is the early stopping time $\delta$, where $R^2/\delta_4$ is larger than $1/\delta^2$. Hence, both the ground truth score $\nabla \log q_t(\cdot)$ and empirical score $\nabla \log \widehat{q}_t(\cdot)$ has Lipschitz constant $R^2/\sigma_t^4$.

# D   AUXILIARY LEMMAS

At the beginning of this part, we provide the previous results of the control of $\nabla \log q_t(\cdot)$ and $\nabla^2 \log q_t(\cdot)$. We note that the proof sketch is almost the same with Benton et al. (2024) (VPSDE), and we provide this part for completeness.

**Lemma 1.** *[The VESDE version of Lemma 5.(Benton et al., 2024)] Considering VESDE forward process Eq. (1), for all $t \geq 0$ and $\forall X_t \in \mathbb{R}^d$, we have that*

$$\nabla \log q_t(X_t) = \frac{\mathbf{m}_t - X_t}{\sigma_t^2} \text{ and } \nabla^2 \log q_t(X_t) = -\sigma_t^{-2} I_d + \sigma_t^{-4}\mathbf{\Sigma}_t.$$

**Proof.** For the first part, we have that

$$\nabla \log q_t(X_t) = \frac{1}{q_t(X_t)}\int_{\mathbb{R}^d}\nabla \log q_{t|0}(X_t|X_0)\, q_{0,t}(X_0, X_t)\, \mathrm{d}X_0.$$

Due to the VESDE forward process, we have that $q_{t|0}(X_t|X_0) = \mathcal{N}\left(X_t; X_0, \sigma_t^2 I\right)$ and $\nabla \log q_{t|0}(X_t|X_0) = -\sigma_t^{-2}(X_t - X_0)$. Then, we have that

$$\nabla \log q_t(X_t) = \mathbb{E}_{q_{0|t}(\cdot|X_t)}\left[-\sigma_t^{-2}(X_t - X_0)\right]$$
$$= -\sigma_t^{-2}X_t + \sigma_t^{-2}\mathbf{m}_t.$$

∎

For the second term, we have that

$$\nabla^2 \log q_t(X_t)$$
$$= \frac{1}{q_t(X_t)}\int_{\mathbb{R}^d}\nabla^2 \log q_{t|0}(X_t|X_0)\, q_{0,t}(X_0, X_t)\, \mathrm{d}X_0$$
$$+ \frac{1}{q_t(X_t)}\int_{\mathbb{R}^d}\left(\nabla \log q_{t|0}(X_t|X_0)\left(\nabla \log q_{t|0}(X_t|X_0)\right)^T q_{0,t}(X_0, X_t)\, \mathrm{d}X_0\right.$$
$$- \frac{1}{q_t(X_t^2)}\left(\int_{\mathbb{R}^d}\nabla \log q_{t|0}(X_t|X_0)\, q_{0,t}(X_0, X_t)\, \mathrm{d}X_0\right)\left(\int_{\mathbb{R}^d}\nabla \log q_{t|0}(X_t|X_0)\, q_{0,t}(X_0, X_t)\, \mathrm{d}X_0\right)^T$$
$$= -\frac{1}{\sigma_t^2}I + \mathbb{E}_{q_{0|t}(\cdot|X_t)}\left[\sigma_t^{-4}(X_t - X_0)(X_t - X_0)^T\right]$$
$$- \mathbb{E}_{q_{0|t}(\cdot|X_t)}\left[-\sigma_t^{-2}(X_t - X_0)\right]\mathbb{E}_{q_{0|t}(\cdot|X_t)}\left[-\sigma_t^{-2}(X_t - X_0)\right]^T$$
$$= -\sigma_t^{-2}I + \sigma_t^{-4}\mathrm{Cov}_{q_{0|t}(\cdot|X_t)}(X_t - X_0)$$
$$= -\sigma_t^{-2}I + \sigma_t^{-4}\mathbf{\Sigma}_t.$$

In the rest of this section, similar to Chen et al. (2023a), we provide a more refined control on the Hessian Matrix $\nabla^2 \log q_t(X_t)$, where $X_t \sim q_t$ instead of the uniform bound. These auxiliary lemmas

are useful for the aggressive exponential decay stepsize. The following two lemmas are exactly the same compared to Chen et al. (2023a), and these lemmas can be used in the VESDE setting since these lemmas do not involve the specific process.

**Lemma 4.** *Let $Q$ be a probability measure on $\mathbb{R}^d$. Consider the density its Gaussian perturbation $q_\sigma(x) \propto \int_{\mathbb{R}^d} \exp\left(-\frac{\|x-y\|^2}{2\sigma^2}\right) \mathrm{d}Q(y)$. Then for $x \sim q_\sigma$, we have the sub-exponential norm bound*

$$\left\|\nabla^2 \log q_\sigma(x)\right\|_{F,\psi_1} \lesssim \frac{d}{\sigma^2},$$

*and*

$$\left\|\nabla \log q_\sigma(x)\right\|_{\psi_2} \lesssim \sqrt{\frac{d}{\sigma^2}}.$$

*where $\|\cdot\|_{F,\psi_1} = \|\|\cdot\|_F\|_{\psi_1}$ denote the sub-exponential norm of the Frobenius norm of a random matrix.*

Let $0 \le t \le s \le T$, the VESDE with $\sigma_t^2 = t^2$ indicates $X_s|X_t \sim \mathcal{N}\left(\alpha_{t,s}X_t, (\sigma_s^2 - \sigma_t^2)I_d\right)$, where $\alpha_{t,s} = 1$. Then, we directly use Lemma 11 of Chen et al. (2023a) to control the discretization error of ground truth socre function.

**Lemma 5.** *For any $0 \le t \le s \le T$, the forward process Eq. (1) satisfies*

$$\mathbb{E}\left\|\nabla \log q_t\left(X_t\right) - \nabla \log q_s\left(X_s\right)\right\|^2$$
$$\le 4\mathbb{E}\left\|\nabla \log q_t\left(X_t\right) - \nabla \log q_t\left(\alpha_{t,s}^{-1}X_s\right)\right\|^2 + 2\mathbb{E}\left\|\nabla \log q_t\left(X_t\right)\right\|^2\left(1 - \alpha_{t,s}^{-1}\right)^2$$

When considering VESDE with $\sigma_t^2 = t^2$, $\alpha_{t,s} = 1$, the above lemma indicates the discretization error is controlled by the space discretization error. Hence, we control the space discretization error of ground truth score function. The following lemma is almost identical compared to Lemma 13 of Chen et al. (2023a) (choosing $\alpha_{t,s} = 1$ since the VESDE setting) except the order of the diffusion (variance) term and the relationship between the variable stepsize and the variance term. For the sake of completeness, we also give the proof process of this part.

**Lemma 6.** *For $0 \le t \le s \le T$ with $\frac{\sigma_s^2 - \sigma_t^2}{\sigma_t^2} \le \frac{1}{2d}$ and the forward process Eq. (1), we have*

$$\mathbb{E}\left\|\nabla \log q_t\left(X_t\right) - \nabla \log q_t\left(X_s\right)\right\|^2 \lesssim \frac{d^2(\sigma_s^2 - \sigma_t^2)}{\sigma_t^4}.$$

**Proof**. To bound the above term by using the Hessian matrix, we have that

$$\nabla \log q_t\left(X_t\right) - \nabla \log q_t\left(X_s\right) = \int_0^1 \nabla^2 \log q_t\left(X_t + a\left(X_s - X_t\right)\right)\left(X_s - X_t\right)\mathrm{d}a$$

$$\mathbb{E}\left\|\nabla \log q_t\left(X_t\right) - \nabla \log q_t\left(X_s\right)\right\|^2 \le \int_0^1 \mathbb{E}\left\|\nabla^2 \log q_t\left(X_t + az_{t,s}\right)z_{t,s}\right\|^2 \mathrm{d}a,$$

where $z_{t,s}$ is defined by $z_{t,s} = X_s - X_t \sim \mathcal{N}\left(0, (\sigma_s^2 - \sigma_t^2)I_d\right)$ and is independent of $X_t$. For random vectors $X, Y$, we use $P_{X,Y}$ to denote the joint probability measure of $(X, Y)$ and $P_{X|Y}$ to denote the conditional probability measure of $X$ given $Y$. Then for $0 \le a \le 1$, we use change of measure to bound $\mathbb{E}\left\|\nabla^2 \log q_t\left(X_t + az_{t,s}\right)z_{t,s}\right\|^2$:

$$\mathbb{E}\left\|\nabla^2 \log q_t\left(X_t + az_{t,s}\right)z_{t,s}\right\|^2 = \mathbb{E}\left[\left\|\nabla^2 \log q_t\left(X_t\right)z_{t,s}\right\|^2 \frac{\mathrm{d}P_{X_t+az_{t,s},z_{t,s}}\left(X_t, z_{t,s}\right)}{\mathrm{d}P_{X_t,z_{t,s}}\left(X_t, z_{t,s}\right)}\right]$$

$$\lesssim \sqrt{\mathbb{E}\left\|\nabla^2 \log q_t\left(X_t\right)z_{t,s}\right\|^4 \mathbb{E}\left(\frac{\mathrm{d}P_{X_t+az_{t,s},z_{t,s}}\left(X_t, z_{t,s}\right)}{\mathrm{d}P_{X_t,z_{t,s}}\left(X_t, z_{t,s}\right)}\right)^2}.$$

Similar to Chen et al. (2023a), we define $M_t = \nabla^2 \log q_t\left(X_t\right)\left(\nabla^2 \log q_t\left(X_t\right)\right)^\top$, $Z_{t,s} = z_{t,s}z_{t,s}^\top$. For $A, B \in \mathbb{R}^{d \times d}$, define the tensor product $A \otimes B \in \left(\mathbb{R}^d\right)^{\otimes 4}$ as $(A \otimes B)_{i_1,i_2,i_3,i_4} = A_{i_1i_2}B_{i_3i_4}$.

Since $M_t$ and $Z_{t,s}$ are independent, then we can bound the two terms in the above inequality separately

$$\mathbb{E} \left\| \nabla^2 \log q_t \left( X_t \right) z_{t,s} \right\|^4 = \langle \mathbb{E} M_t \otimes M_t, \mathbb{E} Z_{t,s} \otimes Z_{t,s} \rangle .$$

Term $\mathbb{E} Z_{t,s} \otimes Z_{t,s}$ is purely determined by the diffusion (variance) term:

$$\mathbb{E} \left( Z_{t,s} \otimes Z_{t,s} \right)_{i_1,i_2,i_3,i_4} = \begin{cases} 3(\sigma_s^2 - \sigma_t^2)^2, & i_1 = i_2 = i_3 = i_4, \\ (\sigma_s^2 - \sigma_t^2)^2, & i_1 \neq i_2, (i_1,i_2) = (i_3,i_4) \text{ or } (i_1,i_2) = (i_4,i_3), \\ 0, & \text{else.} \end{cases}$$

Then, we can bound the inner product term by using exactly the same process compared to Chen et al. (2023a):

$$\langle \mathbb{E} M_t \otimes M_t, \mathbb{E} Z_{t,s} \otimes Z_{t,s} \rangle \lesssim \sigma_{s-t}^4 \left( \sum_{(i_1,i_2)=(i_3,i_4)} + \sum_{(i_1,i_2)=(i_4,i_3)} \right) \mathbb{E} \left( M_t \otimes M_t \right)_{i_1,i_2,i_3,i_4}$$

$$\lesssim (\sigma_s^2 - \sigma_t^2)^2 \mathbb{E} \left\| \nabla^2 \log q_t \left( X_t \right) \right\|_F^4$$

$$\lesssim (\sigma_s^2 - \sigma_t^2)^2 \left( \frac{d}{\sigma_t^2} \right)^4$$

For the rest term, we have that

$$\mathbb{E} \left( \frac{\mathrm{d} P_{X_t + a z_{t,s}, z_{t,s}} \left( X_t, z_{t,s} \right)}{\mathrm{d} P_{X_t, z_{t,s}} \left( X_t, z_{t,s} \right)} \right)^2 = \mathbb{E} \left( \frac{\mathrm{d} P_{X_t + a z_{t,s} | z_{t,s}} \left( X_t | z_{t,s} \right)}{\mathrm{d} P_{X_t | z_{t,s}} \left( X_t | z_{t,s} \right)} \right)^2$$

$$\leq \mathbb{E} \left( \frac{\mathrm{d} P_{X_t + a z_{t,s} | z_{t,s}, x_0} \left( X_t | z_{t,s}, x_0 \right)}{\mathrm{d} P_{X_t | x_0} \left( X_t | x_0 \right)} \right)^2 .$$

We also know that $X_t + a z_{t,s} | (z_{t,s}, x_0) \sim \mathcal{N} \left( x_0 + a z_{t,s}, \sigma_t^2 I_d \right)$ and $X_t | x_0 \sim \mathcal{N} \left( x_0, \sigma_t^2 I_d \right)$. Then, the above term has the following equation by the chi-squared divergence explicitly:

$$\mathbb{E} \left( \frac{\mathrm{d} P_{X_t + a z_{t,s} | z_{t,s}, x_0} \left( X_t | z_{t,s}, x_0 \right)}{\mathrm{d} P_{X_t | x_0} \left( X_t | x_0 \right)} \right)^2 = \mathbb{E} \exp \left( \frac{a^2 \left\| z_{t,s} \right\|^2}{\sigma_t^2} \right) .$$

Recall that we assume $\frac{\sigma_s^2 - \sigma_t^2}{\sigma_t^2} \leq \frac{1}{2d}$, we have that

$$\mathbb{E} \exp \left( \frac{a^2 \left\| z_{t,s} \right\|^2}{\sigma_t^2} \right) = \left( 1 - 2 \frac{a^2 \left( \sigma_s^2 - \sigma_t^2 \right)}{\sigma_t^2} \right)^{-d/2} \lesssim 1 ,$$

and

$$\mathbb{E} \left\| \nabla^2 \log q_t \left( X_t + a z_{t,s} \right) z_{t,s} \right\|^2 \lesssim \frac{d^2 (\sigma_s^2 - \sigma_t^2)}{\sigma_t^4} .$$

Then, we complete the proof. ∎

**Lemma 7.** *Suppose **Assumption 1** holds. Let $\epsilon_{W_2} > 0$. (1) If considering VESDE with $\sigma_t^2 = t^2$, we choose the early stopping parameter $\delta = \frac{\epsilon_{W_2}}{\sqrt{d}}$, (2) If consider VPSDE, we choose $\delta = \epsilon_{W_2}^2 / (\sqrt{d}(R \vee \sqrt{d}))$ then we have $W_2 \left( q_\delta, q_0 \right) \leq \epsilon_{W_2}$.*

**Proof.** For the VESDE forward process Eq. (1), we know that $X_t := X_0 + \sigma_t z$, where $z \sim \text{normal} \left( 0, I_d \right)$ is independent of $X_0$. Hence, for $\delta \lesssim 1$,

$$W_2^2 \left( q_0, q_\delta \right) \leq \mathbb{E} \left[ \left\| \sigma_\delta z \right\|^2 \right] = \sigma_\delta^2 d .$$

Then, for the setting $\sigma_t^2 = t^2$, we can take $\delta \leq \frac{\epsilon_{W_2}}{\sqrt{d}}$.

For the VPSDE forward process, we directly use Lemma 20 of Chen et al. (2022b) to obtain the final results.

∎

