# OpenReview forum: "The Discretization Complexity Analysis of Consistency Models under Variance Exploding Forward Process"
_ICLR.cc/2025/Conference — Submitted to ICLR 2025_

### Official Review · Reviewer_ts5o · 2024-10-30

**Soundness:** 2
**Presentation:** 1
**Contribution:** 3
**Rating:** 5
**Confidence:** 2

**Summary:**

This work studies the discretization complexity of consistency models trained from pre-trained score function, used in practice to reduce the computational burden of diffusion models that require many steps to generate samples.

**Strengths:**

* This work provides the first analysis of consistency distillation, under VESDE.
* Their complexity bound seems reasonable, compared with similar analysis conducted under difference assumption (on the reverse SDE).

**Weaknesses:**

* I think that the manuscript could benefit from organization of the technical terms, for example by introducing an entire section with all the terms and definitions used in this analysis. Currently, many parts of the paper introduce their notation, which makes it hard to follow.
* Minor: Table 1 is hard to read, and I believe captions should be above and not below the table.
* I would suggest that the authors would justify their assumptions and explain why these are reasonable assumption. In the current manuscript, the authors simply state other papers in which these assumptions were made (assumptions 3 and 4).
* The authors should refrain from evaluating their own work (remark 2: "our great results")

**Questions:**

* the authors claim that they "why consistency models have competitive performance compared to diffusion models in application". I don't quite follow why this is the case? The complexity bound hold under certain assumptions that may not hold in practice. Can the authors please clarify this point?

---

> ### Author Response · Authors · 2024-11-19
> **Part 1: the notation and assumption**
>
> Thank you for your valuable comments and suggestions. We provide our response to each question below.  **Please click the PDF  to view our revision paper.**
>
> **Weakness 1: The notation.**
>
> Thanks again for the helpful comments. We have reorganized the notation of this paper in a paragraph (including the definition of discretization complexity and the notation for diffusion and consistency models). Please read **line 298-314** in the revision of our paper (highlighted in green).
>
>
>
> **Weakness 2: The table.**
>
> We have polished the caption of Table 1 and put it above the table. Please read **line 108-113** in the revision of our paper (highlighted in green).
>
> **Weakness 3 (a): The explanation and proof of Assumption 4**
>
> In this part, by calculating the **exact form** of $\nabla \log q\_t(\cdot)$ and $\nabla^2 \log q\_t(\cdot)$}, we show the reason why Assumption 4 matches the true order of $\nabla \boldsymbol{f}^{\text{ex}}$. Let  $\mathbf{m}\_t(X\_t):=\mathbb{E}\_{q\_{0 \mid t}\left(\cdot \mid X\_t\right)}\left[X\_0\right]$ and $\boldsymbol{\Sigma}\_t(X\_t):=\operatorname{Cov}\_{q\_{0 \mid t}\left(\cdot \mid \mathbf{x}_t\right)}\left(X_0\right)$ be the posterior mean and variance. We have the following results.
>
> **Lemma 1.** [The VESDE version of Lemma 5 [1]] Considering the VESDE forward process, for all $t\ge 0$ and $\forall X\_t\in \mathbb{R}^d$, we have that
> $$
> \nabla \log q\_t(X\_t)=\frac{\mathbf{m}\_t-X\_t}{\sigma\_t^{2}} \text{ and } \nabla^2 \log q\_t\left(X\_t\right)=-\sigma\_t^{-2} I\_d+\sigma\_t^{-4} \boldsymbol{\Sigma}\_t\,.
> $$
> It is clear that $\mathbf{m}\_t$ directly maps the noised data to the clean target data distribution, which is exactly $\boldsymbol{f}^{\text{ex}}$ in our work.  Then, the ground-truth score function can be parameterized as $\nabla \log q\_{T-t^{\prime}}(Y,T-t^{\prime})= (\boldsymbol{f}^{\text{ex}}(Y,t^{\prime})-Y)/\sigma\_{T-t^{\prime}}^2$. Using the above parametrization and the second result of Lemma 1, we know that
>
> $$
> \nabla^2\log q\_{T-t^{\prime}}(Y,T-t^{\prime}) = \frac{\nabla \boldsymbol{f}^{\text{ex}}(Y,t^{\prime})-I_d}{\sigma\_{T-t^{\prime}}^2} =-\sigma\_{T-t^{\prime}}^{-2} I_d+\sigma\_{T-t^{\prime}}^{-4} \boldsymbol{\Sigma}\_{T-t^{\prime}}\,.
> $$
> With the above equality, we know that $\nabla \boldsymbol{f}^{\text{ex}}(Y,t^{\prime})=\boldsymbol{\Sigma}\_{T-t^{\prime}}/\sigma\_{T-t^{\prime}}^2$. With the bounded support assumption, we know that $\\|\boldsymbol{\Sigma}\_{T-t^{\prime}}\\|_{op}\leq R^2$. Hence, Assumption 4 matches the true order of $\nabla \boldsymbol{f}^{\text{ex}}$.
>
> We have now added the discussion of the Lipschitz constant in **line 349-375** in the revision of our paper (highlighted in green).
>
> **Weakness 3 (b): The explanation and discussion of Assumption 2\&3.**
>
> Since the form of Assumption 3 is the same compared with the consistency model objective function  $\mathcal{L}\_{\mathrm{CD}}^K$, Assumption 3 means the loss in the training phase is small. Furthermore, the $\left(t_{k+1}^{\prime}-t_k^{\prime}\right)^2$ term matches the $h^2$ order of the discretization error of one-step PFODE process. Hence, our Assumption 3 is reasonable.
>
> For Assumption 2, this term is contained in the score-matching objective function. Hence, this assumption requires the score-matching loss to be small enough in the training phase.
>
> We note that the training process in the application usually guarantees the loss (objective function) converges to a small term. Hence, Assumptions 2 and 3 are realistic assumptions.
>
> **Weakness 4: Refrain from evaluating their own work.**
>
> Thanks again for the helpful comments.  We have polished the presentation of Remark 2 (line 382, highlighted in green).

---

> > ### Author Response · Authors · 2024-11-19
> > **Part 2: Why we explain the empirical success of consistency models from the theoretical perspective**
> >
> > **Question 1. Why we explain the empirical success of consistency models from the theoretical perspective.**
> >
> > The goal of our work is to achieve the competitive discretization complexity compared with diffusion models under the mild assumption and realistic setting (VESDE forward process and EDM stepsize), which is satisfied and used in applications. Then, we say that "we explain the great performance of consistency models from the theoretical perspective".
> >
> > Assumptions 2, 3 and 4 have been discussed in the above part.
> >
> > The bounded support assumption is naturally satisfied by the image dataset and is widely used in theoretical analysis. Furthermore, the bounded support assumption allows the blow-up phenomenon (a widely observed phenomenon in application [2]) of the score function at the end of the reverse process ($t^{\prime}=T-\delta$ or $t=\delta$).
> >
> > Though the bounded support assumption is natural, we still consider some special **unbounded** distribution and show that $L_{f,0}$ still has $1/\sigma_T^2$ order at the beginning of the reverse process (this is crucial for our analysis, line 691 of our revision paper), which indicates our analysis still hold for these unbounded distributions. We also give some intuition as to why it is possible for this special distribution to achieve a better discretization complexity by using the property of these special distributions.
> >
> > *  We first consider a Gaussian distribution $X_0\sim \mathcal{N}(\mu,\Sigma)$, where the posterior mean $\mathbf{m}\_t(X\_t):=\mathbb{E}\_{q\_{0 \mid t}\left(\cdot \mid X\_t\right)}\left[X\_0\right]$ (can be viewed as $\boldsymbol{f}^{\text{ex}}$ in our work) has an analytical form
> >
> > $$
> > \mathbb{E}\left[X_0|X_t\right]=\mu+\left( \Sigma+\sigma_t^2 I_d\right)^{-1} \Sigma\left(X_t-\mu\right)\,.
> > $$
> >
> > ​        It is clear that $\nabla_{X_t}\mathbb{E}[X_0|X_t]=\left( \Sigma+\sigma_t^2 I_d\right)^{-1} \Sigma$ (the $\nabla \boldsymbol{f}^{\text{ex}}$ term), which has the order $1/\sigma_T^2$ at the beginning of the reverse process ($t^{\prime}=0$ or $t=T$). When at the end of the reverse process, since the existence of $\Sigma$ in $\left( \Sigma+\sigma_t^2 I_d\right)^{-1}$, the score will not blow-up, which avoids the $1/\delta$ dependence in Theorem 1 and leads to a better discretization complexity.
> >
> > * For the multimodal distribution such as Gaussian mixture, the score function is highly nonlinear. Hence, similar to [3], we consider a special 2-mode Gaussian mixture distribution $X_0 \sim 1/2*\mathcal{N}(\mu, \sigma^2I_d)+1/2*\mathcal{N}(-\mu, \sigma^2I_d)$. Under this assumption, the ground-truth score has the following form (Appendix A.2 of [3], we transform it from VP to VESDE setting)
> >   $$
> >   \nabla \log q_t(X_t)=\tanh \left(\frac{\mu^{\top} X_t}{\sigma_t^2+\sigma^2}\right) \frac{\mu}{\sigma_t^2+\sigma^2}-\frac{X_t}{\sigma_t^2+\sigma^2}.
> >   $$
> >   Then, we know that
> >   $$
> >   \nabla^2 \log q_t(X_t)=\left(1-\tanh^2 \left(\frac{\mu^{\top} X_t}{\sigma_t^2+\sigma^2}\right)\right)\frac{\mu^{\top}\mu I_d}{(\sigma_t^2+\sigma^2)^2}-\frac{I_d}{\sigma_t^2}.
> >   $$
> >   At the beginning of the reverse process ($t=T$), the $\tanh$ can be approximated by the linear part (Taylor's theorem). It is clear the first part of $\nabla^2 \log q_t(X_t)$ has at least $1/\sigma_T^4$ order. With the discussion of Lemma 1, $\nabla \boldsymbol{f}^{\text{ex}}$ has $1/\sigma_T^2$ order. When at the end of the reverse process, similar to the Gaussian distribution, the score will not blow up, leading to a better discretization complexity.
> >
> >
> >
> > [1] Benton, J., Bortoli, V. D., Doucet, A., & Deligiannidis, G. (2024). Nearly d-linear convergence bounds for diffusion models via stochastic localization.
> >
> > [2] Kim, D., Shin, S., Song, K., Kang, W., & Moon, I. C. (2021). Soft truncation: A universal training technique of score-based diffusion model for high precision score estimation. *arXiv preprint arXiv:2106.05527*.
> >
> > [3] Shah, K., Chen, S., & Klivans, A. (2023). Learning mixtures of gaussians using the DDPM objective. *Advances in Neural Information Processing Systems*, *36*, 19636-19649.

---

> > > ### Comment · Reviewer_ts5o · 2024-11-21
> > > **further comments on quality of writing**
> > >
> > > I appreciate the author's efforts in making some changes and adding a notation section.
> > > However, in its current form, I still find the manuscript hard to follow.
> > > Specifically, the notation section does not provide all the relevant terms, for example the definitions of $q_0$, $s_\phi$, $\hat{Y}^\phi_{t_{k}}$, $\boldsymbol{f}_{\theta}$ among other terms are not provided in this section. For a notation heavy paper, I think it is important to have a clear description of the technical terms, so one does not have to frequently go back and forth between sections.
> > >
> > > I do acknowledge that the assumptions made by the authors seem realistic and that the result is potentially interesting (**even though I am not an expert in this field and hence my low confidence score**), and I therefore increase my rating to 5.
> > >
> > > Minor: typo in line 335, it should be the score matching function that is close enough, not the consistency function.

---

> > > > ### Author Response · Authors · 2024-11-25
> > > >
> > > > Thanks for your valuable comments and positive feedback! We will add a more detailed notation part to our main content and appendix.
> > > >
> > > > We are also happy to answer any further questions.
> > > >
> > > > Best regards,
> > > >
> > > > The Authors

---

### Official Review · Reviewer_J5QF · 2024-10-31

**Soundness:** 2
**Presentation:** 3
**Contribution:** 2
**Rating:** 3
**Confidence:** 3

**Summary:**

The paper introduces a theoretical framework to analyze the discretization complexity of consistency models using a variance-exploding forward process (VESDE) and the EDM discretization scheme, both common in empirical models. Under these settings, the discretization complexity of consistency models can be significantly reduced, achieving results competitive with state-of-the-art diffusion models. Additionally, it examines a multi-step sampling approach to further improve the complexity, effectively reducing training requirements. The authors argue that their approach bridges the gap between theoretical analysis and practical performance, providing insights into why consistency models perform well empirically.

**Strengths:**

(1) The paper is to provide a theoretical analysis of consistency models under realistic conditions, specifically using the variance-exploding forward process (VESDE) and EDM discretization scheme. This aligns with empirical practice and closes the gap between theory and real-world applications.

(2) By analyzing the time-dependent Lipschitz constant and leveraging the EDM step size, the authors achieve a state-of-the-art discretization complexity that rivals diffusion models. This advancement reduces the computational cost of training and sampling in consistency models, making them more efficient for practical applications.

(3) The paper extends its contributions by analyzing multi-step sampling, showing that it can further reduce discretization complexity compared to one-step sampling. This added flexibility makes the method more adaptable to different use cases and aligns well with empirical observations.

(4) The paper provides a thorough comparison of discretization complexity across multiple methods, including diffusion and consistency models with various discretization schemes. This context helps to illustrate the unique benefits and performance of the proposed approach in relation to existing methods.

**Weaknesses:**

(1) Although the paper makes substantial theoretical contributions, it lacks empirical experiments to validate the practical impact of the proposed theoretical improvements. Real-world experiments could demonstrate the actual effectiveness and robustness of the discretization complexity reductions across various datasets and tasks.

(2) The paper assumes an accurate enough approximated score function and consistency function without examining the practical challenges in achieving these accuracies. Since these assumptions are critical to achieving the proposed complexity bounds, this omission may limit the method’s applicability, especially in settings where training perfect consistency functions is difficult.

(3) While the method improves complexity theoretically, its actual scalability to very high-dimensional or large-scale datasets remains untested. For real-world applications like 3D reconstruction or high-resolution video generation, the computational gains may not fully offset the high resource demands, making scalability uncertain.

(4) The paper’s reliance on parameters such as the EDM step size and Lipschitz constant suggests potential sensitivity to hyperparameter tuning. However, it does not offer extensive guidance on optimizing these parameters in practice, which could hinder reproducibility and performance consistency across different scenarios.

(5) The paper’s assumptions, including time-dependent Lipschitz constants and bounded support, while theoretically sound, may not always hold in diverse, real-world data distributions. Such limitations could restrict the generalizability of the method, especially in multimodal distributions where these assumptions might break down.

(6) The paper focuses on the consistency distillation paradigm, which requires pre-trained score functions. This reliance on pre-trained models may limit the standalone applicability of the approach, as obtaining high-quality score functions is often computationally intensive and may not be feasible for all applications.

**Questions:**

This paper investigated consistency models with variance exploding forward process and gradually decay discretization step size, explaining the empirical success of consistency models and achieve the state-of-the-art discretization complexity for consistency models. After obtaining the results of the one-step sampling method of consistency models, they further analyze an existing multi-step consistency sampling algorithm and show that this algorithm improves the discretization complexity compared with one-step generation. The achieved discretization complexity consistency models have obtained competitive with the results of diffusion models. The detailed questions are listed below:
-	Could you provide experimental results or empirical evidence demonstrating the practical impact of your theoretical improvements on discretization complexity? Specifically, how do these theoretical gains translate to performance improvements in high-dimensional, real-world applications like image and video generation?
-	Given the critical role of an "accurate enough" score and consistency function in achieving the complexity bounds, how realistic are these assumptions in practice? What techniques or guidelines do you recommend ensuring that these accuracy conditions are met in real-world applications?
-	Can you discuss the scalability of the proposed framework when applied to very large or high-dimensional datasets? Have you tested the framework on computationally intensive tasks like 3D reconstruction, and if so, what performance gains or limitations did you observe?
-	Your method depends on parameters like the EDM step size and the time-dependent Lipschitz constant. How sensitive is the performance of the model to these hyperparameters, and what recommendations can you provide for optimizing them? Would adaptive methods for step size selection be viable here?
-	While EDM is effective in reducing complexity, did you consider alternative discretization schemes? Are there scenarios where EDM may not be the most optimal choice, and could other schemes, such as non-uniform steps with different decay patterns, offer additional benefits?
-	Your analysis relies on assumptions like bounded support and time-dependent Lipschitz constants. How robust is your approach if these assumptions do not strictly hold in a real-world context, particularly in multimodal or noisy datasets? Could your method be adapted to handle more flexible data assumptions?
-	The framework depends on consistency distillation, which requires a pre-trained score function. In applications where pre-trained score functions are unavailable or costly to obtain, is there a way to apply your framework effectively without a pre-trained model, or could you suggest an alternative training strategy?
-	Have you conducted any error analysis to identify cases where the model may misestimate the required discretization complexity or sampling steps? Understanding the types of samples or data conditions that lead to higher error rates could provide valuable insights into the robustness of your approach.
-	Given the theoretical nature of this work, what practical recommendations or guidelines can you provide to facilitate the deployment of your framework in production settings? Specifically, how can practitioners balance the theoretical complexity improvements with computational feasibility in time-sensitive applications?
-	What are the differences and connections between diffusion models and consistency models? In what ways, we need to consider consistency models instead of diffusion models and vice versa, such as generating samples in a single forward pass, which leads to much faster inference times compared to diffusion models?
-	What are the theoretical insights from Theorem 1? How does the generation error upper bound in Theorem 1 change when considering the introduced noise? All detailed proofs need to be moved in appendix and replaced by technical insights summarization.

---

> ### Author Response · Authors · 2024-11-19
> **Part 1: the empirical evidence, Assumptions and EDM parameters**
>
> Thank you for your valuable comments and suggestions.
>
> As a beginning, we first emphasize our goal in this theoretical work. Our work aims to achieve the competitive discretization complexity compared with diffusion models under the mild assumption and realistic setting (VESDE forward process and EDM stepsize), which is satisfied and used in applications. Then, we say that "we explain the great performance of consistency models from the theoretical perspective". In this work, as shown in Table 1, we achieve $1/\epsilon_{W\_2}^4$ discretization complexity, which has the same order as the SOTA complexity result of diffusion models (In our rebuttal, we discuss each assumption in detail.).
>
> We provide our response to each question below. **Please click the PDF to view our revision paper.**
>
> **Weakness 1& Q1: Empirical evidence.**
>
> Thanks again for your valuable comments on the empirical evidence. One core contribution of our work is to analyze consistency models with great performance in the application (VESDE forward process and EDM stepsize). Hence, empirical works with this setting can be our evidence [1] [2].
>
> **Weakness 2& Q2 (Assumption):  The approximated score and consistency function.**
>
> For Assumption 2, this term is contained in the score-matching objective function. Hence, this assumption requires the score-matching loss to be small enough in the training phase.
>
> For Assumption 3, since the form of Assumption 3 is the same as the consistency model objective function  $\mathcal{L}\_{\mathrm{CD}}^K$, it means the loss in the training phase of the consistency function is small. Furthermore, the $\left(t_{k+1}^{\prime}-t_k^{\prime}\right)^2$ term matches the $h^2$ order of the discretization error of one-step PFODE process.
>
> We note that the training process in the application usually guarantees the loss (objective function) converges a small term. Hence, Assumptions 2 and 3 are realistic assumptions.
>
> **Weakness 3& Q3:  The application to the high-dimension dataset.**
>
> In the application, when dealing with high-dimension data, empirical works use the latent consistency models, which learn the consistency function in the latent space [3]. There are many works [4] [5] that have used LCM in  3D generation and can be viewed as empirical evidence of our work.
>
> Let the latent space dimension be $d^{\prime}$, which is significantly smaller than $d$. Our analysis still holds for latent $d^{\prime}$ and avoids the dependence of $d$.
>
> **Weakness 4& Q4 & Q5:  The discussion of EDM parameter.**
>
> Thanks for the comments on the non-uniform stepsize. We note that the EDM stepsize can cover most of current stepsize with **different decay patterns (with different $a$)**. For example,
>
> * When $a=1$, the  EDM stepsize becomes the uniform stepsize and achieves $1/\epsilon_{W\_2}^{5.5}$, which is worse than the results presented in Theorem 1, which also matches the empirical observation (Figure 13 of [6]).
> * When $a\ge 1$, the stepsize becomes a decay stepsize and a larger $a$ means  a larger decay ration. [6] show that $a=7$ (Figure 13 of [6]) achieves great performance, and our analysis also can provide the discretization complexity under this choice (our revision paper (line 399-400)).
> * When $a\rightarrow +\infty$, the EDM stepsize becomes the theoretical friendly exponential-decay stepsize and a $1/\epsilon\_{W\_2}^4$ result.
>
> For the adaptive stepsize, while it has been used in empirical work [7], it has not been analyzed from the theoretical perspective, and our EDM stepsize covers almost all stepsize choices (since the previous works focus on two extremes: the uniform and exponential-decay). We leave the analysis for the adaptive stepsize as an interesting future work.
>
> **Weakness 4& Q4  (Assumption):  The discussion on Lipschitz constant.**
>
> For the Lipschitz constant of consistency function, as shown in  **line 349-375** in the revision of our paper (highlighted in green), we **prove** that the Lipschitz constant of consistency function matches the true order. Hence, Assumption 4 is reasonable.

---

> ### Author Response · Authors · 2024-11-19
> **Part 2: The bounded support assumption and error analysis**
>
> **Weakness 5 & Q6 (Assumption):  The bounded support assumption and unbounded distribution.**
>
> We note that the bounded support assumption is naturally satisfied by the image dataset, which indicates this assumption allows multimodal distributions. Furthermore, this assumption allows the blow-up phenomenon (a widely observed phenomenon in application) of the score function at the end of the reverse process ($t^{\prime}=T-\delta$ or $t=\delta$). Hence, this assumption is always viewed as a weak and realistic assumption in theoretical works [8].
>
> Though the bounded support assumption is natural, we still consider some special **unbounded** distribution and show that $L_{f,0}$ also has $1/\sigma_T^2$ order at the beginning of the reverse process (this is crucial for our analysis, line 691 of our revision paper), which indicates our analysis still hold for these unbounded distributions. We also give some intuition as to why it is possible for this special distribution to achieve a better discretization complexity by using the property of these special distributions.
>
> *  We first consider a Gaussian distribution $X_0\sim \mathcal{N}(\mu,\Sigma)$, where the posterior mean $\mathbf{m}\_t(X\_t):=\mathbb{E}\_{q\_{0 \mid t}\left(\cdot \mid X\_t\right)}\left[X\_0\right]$ (can be viewed as $\boldsymbol{f}^{\text{ex}}$ in our work) has an analytical form
>
> $$
> \mathbb{E}\left[X_0|X_t\right]=\mu+\left( \Sigma+\sigma_t^2 I_d\right)^{-1} \Sigma\left(X_t-\mu\right)\,.
> $$
>
>   It is clear that $\nabla_{X_t}\mathbb{E}[X_0|X_t]=\left( \Sigma+\sigma_t^2 I_d\right)^{-1} \Sigma$ (the $\nabla \boldsymbol{f}^{\text{ex}}$ term), which has the order $1/\sigma_T^2$ at the beginning of the reverse process ($t^{\prime}=0$ or $t=T$). When at the end of the reverse process, since the existence of $\Sigma$ in $\left( \Sigma+\sigma_t^2 I_d\right)^{-1}$, the score will not blow up, which avoids the $1/\delta$ dependence in Theorem 1 and leads to a better discretization complexity.
>
> * For the multimodal distribution such as a Gaussian mixture, the score function is highly nonlinear. Hence, similar to [9], we consider a special 2-mode Gaussian mixture distribution $X_0 \sim 1/2*\mathcal{N}(\mu, \sigma^2I_d)+1/2*\mathcal{N}(-\mu, \sigma^2I_d)$. Under this assumption, the ground-truth score has the following form (Appendix A.2 of [9], we transform it from VP to VESDE setting)
>
> $$
> \nabla \log q_t(X_t)=\tanh \left(\frac{\mu^{\top} X_t}{\sigma_t^2+\sigma^2}\right) \frac{\mu}{\sigma_t^2+\sigma^2}-\frac{X_t}{\sigma_t^2+\sigma^2}.
> $$
>
> Then, we know that
> $$
> \nabla^2 \log q_t(X_t)=\left(1-\tanh^2 \left(\frac{\mu^{\top} X_t}{\sigma_t^2+\sigma^2}\right)\right)\frac{\mu^{\top}\mu I_d}{(\sigma_t^2+\sigma^2)^2}-\frac{I_d}{\sigma_t^2}.
> $$
> At the beginning of the reverse process ($t=T$), the $\tanh$ can be approximated by the linear part (Taylor's theorem). It is clear the first part of $\nabla^2 \log q_t(X_t)$ has at least $1/\sigma_T^4$ order. With the discussion of Lemma 1 (in our revision paper, line 349-375, highlighted in green), $\nabla \boldsymbol{f}^{\text{ex}}$ has $1/\sigma_T^2$ order. When at the end of the reverse process, similar to the Gaussian distribution, the score will not blow up, leading to a better discretization complexity.
>
>
>
> **Weakness 6 & Q7:  The consistency distillation paradigm.**
>
> As discussed in Remark 1 of our work, no existing results analyze the consistency training paradigm (which does not require the pre-trained score function) under a realistic setting with great performance. Hence,  we left it as an interesting future work to explain the empirical success of the consistency training paradigm from the theoretical perspective under the setting in the application.
>
> **Q8\&Q11: The error analysis.**
>
> The goal of this work is to control each error term to guarantee $W_2\left(\boldsymbol{f}_{\boldsymbol{\theta}, 0 \sharp \mathcal{N}}\left(\mathbf{0}, T^2 I_d\right), q_0\right)\leq \epsilon\_{W_2}$. We require each term to be strictly smaller than $\epsilon\_{W_2}$ and then obtain an upper bound for the discretization complexity, which is a standard analysis process in the complexity analysis area. Hence, our analysis does not misestimate the discretization complexity (steps). In other words, if all assumptions are satisfied, using discretization steps $K$ provided in our work, $W\_2\left(\boldsymbol{f}\_{\boldsymbol{\theta}, 0 \sharp \mathcal{N}}\left(\mathbf{0}, T^2 I_d\right), q_0\right)$ must smaller than $\epsilon\_{W_2}$.
>
> However, it is an interesting future work to analyze the lower bound of the discretization complexity and show our upper bound result is nearly optimal.

---

> > ### Author Response · Authors · 2024-11-19
> > **Part 3: The guidance and intuition of our results**
> >
> > **Q9: The comment and guidance in an empirical setting.**
> >
> > As shown in Sec. 4.1 of our work, we give some guidance in choosing $\tau_2$ when using the multi-step sampling method. More specifically, with a well-learned consistency function  $\epsilon\_{\mathrm{cm}}\leq \epsilon\_{W\_2}^{5/4}$, we can choose a larger $\tau_2=T/2$ to achieve a better discretization complexity. When $\epsilon\_{\mathrm{cm}}$ is slightly larger, we choose $\tau_2=\sqrt{T}$ to reduce the requirement of $\epsilon\_{\mathrm{cm}}$. (More details are shown in Case 1,2 in Sec. 4.1)
> >
> > **Q10: the connection between consistency and diffusion models.**
> >
> > Since the consistency model is a one-step generation model, it is much faster than multi-step diffusion models and is useful in time-sensitive settings. However, when requiring high-quality samples, we always need to run the multi-step (50-1000 steps) diffusion model or multi-step sampling algorithm (2-3 times, as shown in Sec. 4.1 of our paper) of consistency models. In this work, for the first time, we show that multi(2)-step sampling algorithms of consistency models can improve the discretization complexity of the one-step consistency model (Sec. 4.1).
> >
> > In line **298-314** of our revision paper, we also show the detailed complexity definition for diffusion and complexity models (highlighted in green).
> >
> > **Q11. The theoretical insights from Theorem 1.**
> >
> > The theoretical insight is that the consistency model balances different error terms, which leads to a competitive complexity result with diffusion models. It is clear that the first and second terms of the convergence guarantee contain $T$. For the first term, we wish $T$ is large enough to guarantee this term is smaller than $\epsilon\_{W\_2}$. However, a large $T$ would introduce a large second term, which leads to a large discretization complexity. **With the EDM stepsize (a suitable $a$ instead of $a=1$) and time-dependent Lipschitz constant (Before Sec. 5), consistency models balance different error terms and achieve competitive complexity results.**
> >
> >
> >
> > [1] Song, Y., Dhariwal, P., Chen, M., & Sutskever, I. (2023). Consistency models. *arXiv preprint arXiv:2303.01469*.
> >
> > [2] Kim, D., Lai, C. H., Liao, W. H., Murata, N., Takida, Y., Uesaka, T., ... & Ermon, S. (2023). Consistency trajectory models: Learning probability flow ode trajectory of diffusion. *arXiv preprint arXiv:2310.02279*.
> >
> > [3] Luo, S., Tan, Y., Huang, L., Li, J., & Zhao, H. (2023). Latent consistency models: Synthesizing high-resolution images with few-step inference. *arXiv preprint arXiv:2310.04378*.
> >
> > [4] Zhong, Y., Zhang, X., Zhao, Y., & Wei, Y. (2024, October). DreamLCM: Towards High Quality Text-to-3D Generation via Latent Consistency Model. In *Proceedings of the 32nd ACM International Conference on Multimedia* (pp. 1731-1740).
> >
> > [5] Wang, T., Obukhov, A., & Schindler, K. (2024). Consistency^ 2: Consistent and Fast 3D Painting with Latent Consistency Models. *arXiv preprint arXiv:2406.11202*.
> >
> > [6] Karras, T., Aittala, M., Aila, T., & Laine, S. (2022). Elucidating the design space of diffusion-based generative models. *Advances in neural information processing systems*, *35*, 26565-26577.
> >
> > [7] Zhang, H., Wu, Z., Xing, Z., Shao, J., & Jiang, Y. G. (2023). Adadiff: Adaptive step selection for fast diffusion. *arXiv preprint arXiv:2311.14768*.
> >
> > [8] De Bortoli, V. (2022). Convergence of denoising diffusion models under the manifold hypothesis. *arXiv preprint arXiv:2208.05314*.
> >
> > [9] Shah, K., Chen, S., & Klivans, A. (2023). Learning mixtures of gaussians using the DDPM objective. *Advances in Neural Information Processing Systems*, *36*, 19636-19649.

---

> > > ### Comment · Reviewer_J5QF · 2024-11-26
> > > **Further discussion**
> > >
> > > I appreciate the authors' feedback regarding the questions about the manuscript. However, in its current version, some key questions are still not answered well, for example, How does the generation error upper bound in Theorem 1 change when considering the introduced noise? In what ways, we need to consider consistency models instead of diffusion models and vice versa, such as generating samples in a single forward pass, which leads to much faster inference times compared to diffusion models?
> > > - Although the paper makes substantial theoretical contributions, it lacks empirical experiments to validate the practical impact of the proposed theoretical improvements. Real-world experiments could demonstrate the actual effectiveness and robustness of the discretization complexity reductions across various datasets and tasks.
> > > - The paper assumes an accurate enough approximated score function and consistency function without examining the practical challenges in achieving these accuracies. Since these assumptions are critical to achieving the proposed complexity bounds, this omission may limit the method’s applicability, especially in settings where training perfect consistency functions is difficult.
> > > - While the method improves complexity theoretically, its actual scalability to very high-dimensional or large-scale datasets remains untested.
> > > - The paper’s reliance on parameters such as the EDM step size and Lipschitz constant suggests potential sensitivity to hyperparameter tuning. However, it does not offer extensive guidance on optimizing these parameters in practice, which could hinder reproducibility and performance consistency across different scenarios.
> > > - The paper’s assumptions, including time-dependent Lipschitz constants and bounded support, while theoretically sound, may not always hold in diverse, real-world data distributions. Such limitations could restrict the generalizability of the method, especially in multimodal distributions where these assumptions might break down.
> > > - The paper focuses on the consistency distillation paradigm, which requires pre-trained score functions. This reliance on pre-trained models may limit the standalone applicability of the approach, as obtaining high-quality score functions is often computationally intensive and may not be feasible for all applications.

---

> > > > ### Comment · Reviewer_J5QF · 2024-12-01
> > > > **Discussion**
> > > >
> > > > Thank you for the authors' response. Due to the above concerns, I would maintain my score.

---

> > > > > ### Author Response · Authors · 2024-12-03
> > > > >
> > > > > Thanks for your valuable comments and further feedback.

---

> ### Author Response · Authors · 2024-11-21
> **Part4: Further discussion on adaptive and EDM stepsize**
>
> Dear reviewer,
>
> Thanks again for your comments on adaptive stepsize and EDM stepsize. In response to Weakness 4&Q4 &5, we show that EDM stepsize covers most of the current stepsize with different decay patterns. In this part, we further discuss the difference between adaptive stepsize and EDM stepsize and show that the current adaptive stepsize relies on a trained NN [7]. On the contrary, EDM stepsize is widely used in empirical works [6] [10] and does not rely on a trained NN.
>
> For the first point, to determine the adaptive stepsize for different prompts, [7] train an additional NN by using the reinforcement learning method, which would introduce additional training costs. As shown in Section 4 of [6] ([6] use EDM stepsize with $a=7$), it is an open question if adaptive solvers can be a net win over a well-tuned fixed schedule in sampling diffusion models. Currently, the empirical work still adopts EDM stepsize and achieves great performance [10].
>
>
>
> [10] Karras, T., Aittala, M., Lehtinen, J., Hellsten, J., Aila, T., & Laine, S. (2024). Analyzing and improving the training dynamics of diffusion models. In *Proceedings of the IEEE/CVF Conference on Computer Vision and Pattern Recognition* (pp. 24174-24184).

---

> > ### Author Response · Authors · 2024-11-25
> >
> > Dear Reviewer  J5QF,
> >
> > Thank you once again for dedicating your valuable time to reviewing our paper and considering our responses. As the author-reviewer discussion period is nearing its conclusion, we would greatly appreciate it if you could let us know whether you have any remaining concerns.
> >
> > Best regards,
> >
> > The Authors

---

### Official Review · Reviewer_Mmdg · 2024-11-03

**Soundness:** 3
**Presentation:** 3
**Contribution:** 3
**Rating:** 6
**Confidence:** 1

**Summary:**

This paper differentiates itself by bridging the gap between theory and practice in consistency models, which hasn’t been fully addressed in previous research. While earlier works often relied on a variance-preserving forward process (VPSDE) and uniform step sizes for noise injection, these setups don’t align with the high-performing configurations used in practice. Real-world consistency models instead use a variance-exploding forward process (VESDE) and an EDM (exponential decay model) discretization scheme, where the step size starts large and gradually decreases, making sampling more efficient and accurate. By focusing on these practical settings, this paper provides a theoretical foundation that explains the empirical success of consistency models, shedding light on why these design choices lead to strong results in real applications.

**Strengths:**

The paper provides a well-written background on consistency models, highlighting what differentiates them from diffusion models and clearly explaining the motivation for this study—namely, that previous theoretical work does not align with practical design choices that lead to optimal performance. The supplementary materials offer essential background information, enhancing the paper’s completeness.

**Weaknesses:**

Although the paper is theoretical in nature, it includes several empirical claims—such as the assertion that previous work does not operate in a realistic setting, or that the results are superior to those of other generative models and theoretical approaches. These claims should ideally be supported within the paper by empirical evidence to strengthen their validity.

**Questions:**

What is defined as great performance? that term is repreated multiple times.

---

> ### Author Response · Authors · 2024-11-19
> **The empirical evidence and the definitiaon of great performance**
>
> Thank you for your valuable comments and suggestions. We provide our response to each question below.  **Please click the PDF to view our revision paper.**
>
> **Weakness 1: Empirical evidence.**
>
> Thanks again for your valuable comments on the empirical evidence. One core contribution of our work is to analyze consistency models with great performance (VESDE forward process and EDM stepsize). Hence, empirical works with this setting can be our evidence [1] [2].
>
> **Question 1: The definition of great performance.**
>
> The goal of our work is to achieve the competitive discretization complexity compared with diffusion models under the mild assumption and realistic setting (VESDE forward process and EDM stepsize), which is satisfied and used in applications. Then, we say that "we explain the great performance of consistency models from the theoretical perspective". In this work, as shown in Table 1, we achieve $1/\epsilon_{W\_2}^4$ discretization complexity, which has the same order as the SOTA complexity result of diffusion models.
>
> In the following part, we discuss each assumption.
>
> * Assumption 1: The bounded support assumption is naturally satisfied by the image dataset and is widely used in theoretical analysis. Furthermore, the bounded support assumption allows the blow-up phenomenon (a widely observed phenomenon in application [3]) of the score function at the end of the reverse process ($t^{\prime}=T-\delta$ or $t=\delta$).
>
> * Assumption 2\&3: For Assumption 2, this term is contained in the score-matching objective function. Hence, this assumption requires the score-matching loss to be small enough in the training phase.
>
>   For Assumption 3, since the form of Assumption 3 is the same as the consistency model objective function  $\mathcal{L}\_{\mathrm{CD}}^K$, it means the loss in the training phase is small, which is usually satisfied in the application. Furthermore, the $\left(t_{k+1}^{\prime}-t_k^{\prime}\right)^2$ term matches the $h^2$ order of the discretization error of one-step PFODE process.
>
>   We note that the training process in the application usually guarantees the loss (objective function) converges a small term. Hence, Assumptions 2 and 3 are realistic assumptions.
>
> * For Assumption 4, as shown in  **line 349-375** in the revision of our paper (highlighted in green), the Lipschitz constant of the approximated consistency function matches the true order and is reasonable.
>
>
>
> [1] Song, Y., Dhariwal, P., Chen, M., & Sutskever, I. (2023). Consistency models. *arXiv preprint arXiv:2303.01469*.
>
> [2] Kim, D., Lai, C. H., Liao, W. H., Murata, N., Takida, Y., Uesaka, T., ... & Ermon, S. (2023). Consistency trajectory models: Learning probability flow ode trajectory of diffusion. *arXiv preprint arXiv:2310.02279*.
>
> [3] Kim, D., Shin, S., Song, K., Kang, W., & Moon, I. C. (2021). Soft truncation: A universal training technique of score-based diffusion model for high precision score estimation. *arXiv preprint arXiv:2106.05527*.

---

### Official Review · Reviewer_ahVo · 2024-11-04

**Soundness:** 2
**Presentation:** 3
**Contribution:** 2
**Rating:** 3
**Confidence:** 4

**Summary:**

This paper studies the theoretical aspect of consistency models with VESDE forward process and EDM discretization scheme. The discretization complexity to achieve a small error in Wasserstein distance is studied. Multi-step sampling is proven to improve the discretization complexity.

**Strengths:**

1. This paper studies the consistency model with variance exploding forward process and decaying stepsize, which are both widely used empirically;
2. Multistep sampling as an iterative sampling procedure to reduce error is studied.

**Weaknesses:**

1. The main result heavily depends on $L$, the Lipschitz constant of the exact consistency function $f^{ex}$. According to the equation in line 689 and 701, the reverse beginning error (first term in Line 669) is bounded by $L\cdot R$, where $R$ is the radius of the support of the target distribution. As a result, when $L = \Omega(1)$ the result becomes meaningless since $R$ is the largest error possible. There are some discussions in line 342-348 regarding the $L$, but I'm not sure if I understand them correctly. In particular,
  - could you please provide a detailed derivation to obtain the equation in line 342 using eq 3 of Karras et al.? Why does the exact consistency function minimize the $L2$ error in eq 2?
  - I think the equation in line 345 implies $||\nabla f|| \le 1 + R^2/\sigma^2$. The constant $1$ will make Theorem 1 meaningless. The reverse beginning error is then bounded by $(1 + R^2/\sigma^2) R$ following the steps in line 689 and 701. Since a trivial mapping as simple as $f(x,t) = 0, \forall x,t$ already achieves $W_2 \le R$, the main result becomes meaningless.
  - could you please provide an example on how the current result apply to multimodal distributions, like Bernoulli distribution or Gaussian mixture? A phase transition in the consistency function is anticipated, i.e. a threshold to map the noise to different modes. Will the consistency function have a large Lipschitz constant around such a threshold?

2. Why the analysis was limited to 2 steps? Are there specific technical challenges in extending to more steps? is there a further improvement when sampling with 3 steps?

**Questions:**

Could you provide a step-by-step derivation on the application of Gronwall's inequality in line 874 - 882?

---

> ### Author Response · Authors · 2024-11-19
> **Part 1: The discussion and proof on the Lipschitz constant of consistency function**
>
> Thank you for your valuable comments and suggestions. We provide our response to each question below.  **Please click the PDF to view our revision paper.**
>
> **Weakness 1(b). The order of the Lipschitz constant of consistency function.**
>
> In this part, by calculating the **exact form** of $\nabla \log q\_t(\cdot)$ and $\nabla^2 \log q\_t(\cdot)$}, we show the reason why Assumption 4 matches the true order of $\nabla \boldsymbol{f}^{\text{ex}}$ and would not lead to a meaningless result.
>
> Let  $\mathbf{m}\_t(X\_t):=\mathbb{E}\_{q\_{0 \mid t}\left(\cdot \mid X\_t\right)}\left[X\_0\right]$ and $\boldsymbol{\Sigma}\_t(X\_t):=\operatorname{Cov}\_{q\_{0 \mid t}\left(\cdot \mid \mathbf{x}_t\right)}\left(X_0\right)$ be the posterior mean and variance. We have the following results.
>
> **Lemma 1.** [The VESDE version of Lemma 5 [1]]Considering VESDE forward process, for all $t\ge 0$ and $\forall X\_t\in \mathbb{R}^d$, we have that
> $$
> \nabla \log q\_t(X\_t)=\frac{\mathbf{m}\_t-X\_t}{\sigma\_t^{2}} \text{ and } \nabla^2 \log q\_t\left(X\_t\right)=-\sigma\_t^{-2} I\_d+\sigma\_t^{-4} \boldsymbol{\Sigma}\_t\,.
> $$
> It is clear that $\mathbf{m}\_t$ directly maps the noised data to the clean target data distribution, which is exactly $\boldsymbol{f}^{\text{ex}}$ in our work.  Then, the ground-truth score function can be parameterized as $\nabla \log q\_{T-t^{\prime}}(Y,T-t^{\prime})= (\boldsymbol{f}^{\text{ex}}(Y,t^{\prime})-Y)/\sigma\_{T-t^{\prime}}^2$. Using the above parametrization and the second result of Lemma 1, we know that
> $$
> \nabla^2\log q\_{T-t^{\prime}}(Y,T-t^{\prime}) = \frac{\nabla \boldsymbol{f}^{\text{ex}}(Y,t^{\prime})-I_d}{\sigma\_{T-t^{\prime}}^2} =-\sigma\_{T-t^{\prime}}^{-2} I_d+\sigma\_{T-t^{\prime}}^{-4} \boldsymbol{\Sigma}\_{T-t^{\prime}}\,.
> $$
> With the above equality, we know that $\nabla \boldsymbol{f}^{\text{ex}}(Y,t^{\prime})=\boldsymbol{\Sigma}\_{T-t^{\prime}}/\sigma\_{T-t^{\prime}}^2$. With the bounded support assumption, we know that $\\|\boldsymbol{\Sigma}\_{T-t^{\prime}}\\|_{op}\leq R^2$. Hence, Assumption 4 matches the true order of $\nabla \boldsymbol{f}^{\text{ex}}$.
>
> We have now added the discussion of the Lipschitz constant in **line 349-375** in the revision of our paper (highlighted in green).

---

> > ### Comment · Reviewer_ahVo · 2024-11-24
> >
> > Thank you very much for your clarification. I'm still confused by line 356, which claims that the consistency function can be written as the sum of the scaled score function and identity function. It's not clear to me if this is true. Is it true for $q_0 = N(0,1)$?

---

> > > ### Author Response · Authors · 2024-11-25
> > > **The special standard Gaussian case**
> > >
> > > Thanks very much for the further discussion. In this part, we use a special case $q_0=\mathcal{N}(0,1)$ to show why the consistency function can be written as the sun of the scaled score function and identity function.
> > >
> > > In this case, with the VESDE forward, $q_t= \mathcal{N}(0,1+\sigma_t^2)$, which means the score function has the following form:
> > > $$
> > > \nabla \log q_t(X_t)=\frac{-X_t}{1+\sigma_t^2}\,.
> > > $$
> > > For the consistency function, by using Example 1, we know that (here we use the forward process $t$ and $X_t$ for clarity)
> > > $$
> > > \boldsymbol{f}^{\text{ex}}(X_t,t)=\mathbb{E}\left[X_0|X_t\right]=\left(1+\sigma_t^2 \right)^{-1} X_t.
> > > $$
> > > Since the forward process only changes the variance in this setting, the consistency function only compresses the variance, which matches the intuition. Finally, we provide the calculation to show why Lemma 1 holds.
> > >
> > > Here, we recall the first part of Lemma 1 provided by our revision paper:
> > > $$
> > > \nabla \log q\_t(X\_t)=\frac{\mathbb{E}\left[X_0|X_t\right]-X\_t}{\sigma\_t^{2}}.
> > > $$
> > > By calculating the right hand of Lemma 1, we know that
> > > $$
> > > (\mathbb{E}\left[X_0|X_t\right]-X\_t)/\sigma_t^2=(\frac{X_t}{1+\sigma_t^2}-\frac{(1+\sigma_t^2)X_t}{1+\sigma_t^2})/\sigma_t^2=\frac{-X_t}{1+\sigma_t^2},
> > > $$
> > > which is the ground-truth score function.
> > >
> > > We hope the above clarifications will help to address the concerns of the reviewer.

---

> > > > ### Comment · Reviewer_ahVo · 2024-11-25
> > > >
> > > > How to see that $\left(1+\sigma_t^2 \right)^{-1} X_t$ is the true consistency function? For $X_t \sim q_t = N(0,1+\sigma_t^2)$, we would expect: $f^{ex}(X_t,t) \sim q_0 = N(0,1)$. However, $\left(1+\sigma_t^2 \right)^{-1} X_t \sim N(0, (1+\sigma_t^2)^{-1}) \neq N(0,1)$. So I think $f^{ex}(X_t,t) \neq \left(1+\sigma_t^2 \right)^{-1} X_t$.

---

> ### Author Response · Authors · 2024-11-19
> **Part 2: The special  multimodal unbounded distribution**
>
> **Weakness 1 (c). The order of special Gaussian and Gaussian Mixture cases. (Unbounded distribution)**
>
> Since the bounded support assumption is naturally satisfied by the image dataset, the $R^2/\sigma_t^2$  order for $\nabla \boldsymbol{f}^{\text{ex}}$ holds for a wide range of distribution. Furthermore, the bounded support assumption allows the blow-up phenomenon (a widely observed phenomenon in application [2]) of the score function at the end of the reverse process ($t^{\prime}=T-\delta$ or $t=\delta$).
>
> Though the bounded support assumption is natural, we still consider some special **unbounded** distribution and show that $L_{f,0}$ also has $1/\sigma_T^2$ order at the beginning of the reverse process (this is crucial for our analysis, line 691 of our revision paper). We also give some intuition as to why it is possible for this special distribution to achieve a better discretization complexity by using the property of these special distributions.
>
> *  We first consider a Gaussian distribution $X_0\sim \mathcal{N}(\mu,\Sigma)$, where the posterior mean $\mathbf{m}\_t(X\_t):=\mathbb{E}\_{q\_{0 \mid t}\left(\cdot \mid X\_t\right)}\left[X\_0\right]$ (can be viewed as $\boldsymbol{f}^{\text{ex}}$ in our work) has an analytical form
>
> $$
> \mathbb{E}\left[X_0|X_t\right]=\mu+\left( \Sigma+\sigma_t^2 I_d\right)^{-1} \Sigma\left(X_t-\mu\right)\,.
> $$
>
> ​        It is clear that $\nabla_{X_t}\mathbb{E}[X_0|X_t]=\left( \Sigma+\sigma_t^2 I_d\right)^{-1} \Sigma$ (the $\nabla \boldsymbol{f}^{\text{ex}}$ term), which has the order $1/\sigma_T^2$ at the beginning of the reverse process ($t^{\prime}=0$ or $t=T$). When at the end of the reverse process, since the existence of $\Sigma$ in $\left( \Sigma+\sigma_t^2 I_d\right)^{-1}$, the score will not blow up, which avoids the $1/\delta$ dependence in Theorem 1 and leads to a better discretization complexity.
>
> * For the multimodal distribution such as a Gaussian mixture, the score function is highly nonlinear. Hence, similar to [3], we consider a special 2-mode Gaussian mixture distribution $X_0 \sim 1/2*\mathcal{N}(\mu, \sigma^2I_d)+1/2*\mathcal{N}(-\mu, \sigma^2I_d)$. Under this assumption, the ground-truth score has the following form (Appendix A.2 of [3], we transform it from VP to VESDE setting)
>   $$
>   \nabla \log q_t(X_t)=\tanh \left(\frac{\mu^{\top} X_t}{\sigma_t^2+\sigma^2}\right) \frac{\mu}{\sigma_t^2+\sigma^2}-\frac{X_t}{\sigma_t^2+\sigma^2}.
>   $$
>   Then, we know that
>   $$
>   \nabla^2 \log q_t(X_t)=\left(1-\tanh^2 \left(\frac{\mu^{\top} X_t}{\sigma_t^2+\sigma^2}\right)\right)\frac{\mu^{\top}\mu I_d}{(\sigma_t^2+\sigma^2)^2}-\frac{I_d}{\sigma_t^2}.
>   $$
>   At the beginning of the reverse process ($t=T$), the $tanh$ can be approximated by the linear part (Taylor's theorem). It is clear the first part of $\nabla^2 \log q_t(X_t)$ has at least $1/\sigma_T^4$ order. With the discussion of Lemma 1, $\nabla \boldsymbol{f}^{\text{ex}}$ has $1/\sigma_T^2$ order. When at the end of the reverse process, similar to the Gaussian distribution, the score will not blow up, leading to a better discretization complexity.

---

> > ### Author Response · Authors · 2024-11-19
> > **Part 3: The other step-by-step derivation**
> >
> > **Weakness 1 (a). The link between $\nabla \log q_{T-t^{\prime}}\left(Y, T-t^{\prime}\right)=\left(\boldsymbol{f}^{\mathrm{ex}}\left(Y, t^{\prime}\right)-Y\right) / \sigma_{T-t^{\prime}}^2$ and Eq (3) of [4].**
> >
> > As shown in W1 (b),  $\nabla \log q\_{T-t^{\prime}}\left(Y, T-t^{\prime}\right)=\left(\boldsymbol{f}^{\mathrm{ex}}\left(Y, t^{\prime}\right)-Y\right) / \sigma_{T-t^{\prime}}^2$ is exactly the first equation of the above Lemma 1 since $\boldsymbol{f}^{\mathrm{ex}}$ is equivalent to the posterior mean $\mathbf{m}\_t(X\_t)$, In this part, we discuss the the relationship between Eq (3) of [4] and $\boldsymbol{f}^{\mathrm{ex}}$ of consistency models. We first recall the parametrization of [4] (Eq (3)):
> > $$
> > \nabla \log q_t(X_t)=(D(X_t,t)-X_t)/\sigma_t^2\,.
> > $$
> >
> > * It is clear that $\boldsymbol{f}^{\mathrm{ex}}$ and $D(X_t,t)$ are just different notation of $\mathbf{m}_t[X\_0|X\_t]$ in different paper.
> >
> > * The difference between [4] and consistency models is the training process to obtain the one-step mapping.
> >
> > **The training objective of [4] is equivalent to the score matching.**
> >
> > [4] parameterize the approximated score $s\_{\phi}(X\_t,t)=(D\_{\phi}(X\_t,t)-X\_t)/\sigma_t^2$ and use the following loss function ([4], Eq. 2) to learn $D_{\phi}$:
> > $$
> > \begin{align*}
> >     \min\_{\phi\in \Phi} \int\_0^T  \mathbb{E}\_{X_0}\left[\mathbb{E}\_{X_t|X_0}\left\\|X\_0-D\_{\phi}\left(X\_t, t\right)\right\\|\_2^2\right] \mathrm{d} t\,.
> > \end{align*}
> > $$
> > Since $\nabla \log q_t(X_t|X_0)=\frac{X_0-X_t}{\sigma_t^2}$, the above  objective is equivalent to the standard score matching objective function.
> > $$
> > \begin{align*}
> >     \min\_{\phi\in \Phi} \int\_0^T  \mathbb{E}\_{X_0}\left[\mathbb{E}\_{X_t|X_0}\left\\|\nabla \log q\_t\left(X\_t|X_0\right)-s_{\phi}\left(X_t, t\right)\right\\|\_2^2\right] \mathrm{d} t\,.
> > \end{align*}
> > $$
> > Hence, using the above score matching loss function can not learn a good enough $D_{\phi}$. As a solution,  [4] use $s\_{\phi}(X\_t,t)=(D\_{\phi}(X\_t,t)-X\_t)/\sigma_t^2$ to run the discrete reverse process ($K$ steps) to generate samples (Line 7-9 of Algorithm 2, [4]).
> >
> > **The training paradigm of consistency models achieves consistency function with great performance. **
> >
> > As shown in empirical work [5], the consistency function is obtained by using the consistency objective $\mathcal{L}_{\mathrm{CD}}^K$. In this work, we analyze the discretization complexity of consistency models under this training paradigm (with VESDE forward process and EDM stepsize, which is adopted by consistency models with great empirical performance).
> >
> > **Weakness 2. The multi-step sampling algorithm.**
> >
> > We note our analysis can be directly extended to multi-step sampling. Since [5] show that $2$-step sampling is enough for consistency models (Table 1 [5], the NFE number is the sampling number) to achieve great performance, we use $2$-step sampling as an example for a clearer discussion.
> >
> > We have now added the discussion in **line 437-439** in the revision of our paper (highlighted in green).
> >
> > For $3$-step sampling, we use $\tau_1=T/2$ and $\tau_2=T/4$ as an example.  With this choice, the guarantee becomes (we only consider terms corresponding to $\epsilon$ here)
> > $$
> > W_2(p_3,q_0)\leq \delta+\frac{1}{T^6}+\frac{(T/\delta)^{\frac{1}{2a}}}{\sqrt{K}\delta}\,,
> > $$
> > which would leads to a better $1/\epsilon_{W\_2^{4+\frac{7}{6a}}}$.
> >
> >
> >
> > **Q1: A step-by-step derivation on the application of Gronwall's inequality.**
> >
> > Thanks for the valuable comment. We have now added the step-by-step derivation on the application of Gronwall's inequality in line **864-886** in the revision of our paper (highlighted in green).
> >
> > [1] Benton, J., Bortoli, V. D., Doucet, A., & Deligiannidis, G. (2024). Nearly d-linear convergence bounds for diffusion models via stochastic localization.
> >
> > [2] Kim, D., Shin, S., Song, K., Kang, W., & Moon, I. C. (2021). Soft truncation: A universal training technique of score-based diffusion model for high precision score estimation. *arXiv preprint arXiv:2106.05527*.
> >
> > [3] Shah, K., Chen, S., & Klivans, A. (2023). Learning mixtures of gaussians using the DDPM objective. *Advances in Neural Information Processing Systems*, *36*, 19636-19649.
> >
> > [4] Karras, T., Aittala, M., Aila, T., & Laine, S. (2022). Elucidating the design space of diffusion-based generative models. *Advances in neural information processing systems*, *35*, 26565-26577.
> >
> > [5] Song, Y., Dhariwal, P., Chen, M., & Sutskever, I. (2023). Consistency models. *arXiv preprint arXiv:2303.01469*.

---

> ### Author Response · Authors · 2024-11-25
>
> Thanks for your quick and detailed response. Since $\mathbb{E}\left[X_0|X_t\right]$ is posterior mean, the $\boldsymbol{f}^{\text{ex}}(X_t,t)= \mathbb{E}\left[X_0|X_t\right]$ can generate the ground-truth **mean** for a large $T$. Now, let us consider $q_0=\mathcal{N}(\mu,1)$ (by similar calculation with the above discussion, Lemma 1 still hold). Then, we have that
> $$
> \boldsymbol{f}^{\text{ex}}(X_t,t)=\mathbb{E}\left[X_0|X_t\right]=\mu+\left(1+\sigma_t^2\right)^{-1}\left(X_t-\mu\right)\,.
> $$
> It is clear that $X_t-\mu\sim \mathcal{N}(0,1+\sigma_t^2)$. We note that $(1+\sigma_t^2)^{-1}$ coefficient is the core to generate true mean. More specifically, $\left(1+\sigma_t^2\right)^{-1}\left(X_t-\mu\right)\sim  \mathcal{N}(0,(1+\sigma_t^2)^{-1})$, which would goes to $0$ with a large enough $T$ (In our work, we also choose a large enough $T$) and does not introduce a large influence on the first $\mu$ term.
>
> We also provide the solution of posterior covariance
> $$
> Cov\left[X_0|X_t\right]=1-\frac{1}{1+\sigma_t^2}
> $$
> which is independent of $X_t$ and $\nabla_{X\_t} Cov\left[X_0|X_t\right] = 0$.
>
> Though the good results of posterior covariance (under Gaussian distribution), we note that when considering a more general $q_0$, the posterior covariance maybe depends on $X_t$. However, as shown in [4], they directly view $\mathbb{E}\left[X_0|X_t\right]$ as the ground truth denoised autoencoder (DAE) (the consistency function in our work) and empirical consistency models [5] also follow this setting, we say that $\mathbb{E}\left[X_0|X_t\right]$ is the consistency function in our work.
>
> Thanks again for your vauleabl comments. We will add the above discussion in our revision paper.

---

> > ### Author Response · Authors · 2024-11-25
> > **The discussion on the intuition of EDM and consistency model**
> >
> > In this part, we recall why EDM and consistency model adopt the VESDE forward process ($\sigma_t^2=t^2$) and why it is possible for this process to achieve one-step generation (Sec 3. of [4], paragraph Trajectory curvature and noise schedule). We note that with this forward process, the reverse PFODE (Eq 2 of our paper, $\eta=0$) has the following form  (here we use the forward time $t$ and $X_t$ for clarity. In the forward timeline, the reverse process starts from $X_T$ and goes to $X_0$):
> > $$
> > \begin{align*}
> > \mathrm{d} X_{t}=t \nabla \log q_{t}\left(X_{t}\right) \mathrm{d} t, X_T\sim q_T.
> > \end{align*}
> > $$
> > Due to the lemma 1, we know
> > $$
> > \nabla \log q\_t(X\_t)=\frac{\mathbb{E}\left[X_0|X_t\right]-X\_t}{t^{2}}\,.
> > $$
> > Then, the reverse process become
> > $$
> > \begin{align*}
> > \mathrm{d} X_{t}=\frac{\mathbb{E}\left[X_0|X_t\right]-X_t}{t} \mathrm{d} t,
> > \end{align*}
> > $$
> > which indicates it is possible to do one-step generation (we note the generated samples as $\bar{X}_0$) with the Euler method starting from $X_T$:
> > $$
> > \bar{X}_0= X_T+T\frac{\mathbb{E}\left[X_0|X_T\right]-X_T}{T} = \mathbb{E}\left[X_0|X_T\right].
> > $$
> > Hence, [4] [5] say that the VESDE forward process ($\sigma_t^2=t^2$) has the potential of one-step generation and view $\mathbb{E}\left[X_0|X_t\right]$ as the ground truth denoised autoencoder (DAE) (the consistency function in our work).

---

> > > ### Comment · Reviewer_ahVo · 2024-11-25
> > >
> > > According to the original consistency model paper (Song et al. 2023), the PF-ODE for the VE SDE should be
> > > $$
> > > \frac{d X_t}{dt} = -t \nabla \log q_t(X_t),
> > > $$
> > > so the equation above is missing a minus sign. It's not clear to me how this $\bar X_0$ relates to the solution to this ODE.
> > >
> > > I would suggest the author do the following to check the correctness of line 356 for the simple example when $\sigma_t^2=t^2$ and the data distribution is Gaussian, $q_0=N(0,1)$:
> > > 1. Write down the closed form of the score function and the closed form of  $E[X_0 | X_t]$ using Lemma 1;
> > > 2. Instiantiate the PF-ODE with this score function and find the closed-form solution. By definition, this solution is the true consistency function $f^{ex}$. Then compare this solution to the closed form of $E[X_0 | X_t]$;
> > > 3. Let $f(X, t) = t^2 \nabla\log q_t(X) + X$. By Lemma 1, $f(X_t,t)$ is equal to $E[X_0 | X_t]$. Let $Z \sim q_t = N(0, t^2 + 1)$, then find the distribution of $f(Z,t)$ and compare this to $q_0=N(0,1)$. Note that for $f^{ex}$, the distribution of $f^{ex}(Z,t)$ is $q_0$.

---

> > > > ### Author Response · Authors · 2024-12-03
> > > >
> > > > We thank you for the valuable and helpful discussion with the reviewer. After the calculation, the consistency function for a Gaussian distribution $q_0 =\mathcal{N}(\mu,\sigma^2)$ is $Y_T= \mu + \frac{Y_0-\mu}{\sqrt{\sigma^2+T^2}}\sigma$, which indicates the Lipschitz constant of consistency function (of Gaussian distribution) at the beginning of the reverse process is $1/T$ (not currently $1/T^2$).
> > > >
> > > > Then, we can assume that the Lipschitz constant of the consistency function at the reverse beginning is $1/T$. The intuition is that we need a $1/T$ to remove the influence of larger $\sigma_T^2=T^2$ when mapping to the target distribution. We also do simulation experiments on highly non-log-concave GMM distribution to verify and support our intuition. We also note that [1] also show similar observations (Eq. (7) of their work).
> > > >
> > > > Hence, a reasonable assumption is that (a) $L_{f,0}=1/T$ (as in the above discussion) and (2) the Lipschitz constant of consistency function is uniformly bounded by $L\_f$ (a standard assumption of previous works).
> > > >
> > > > With this assumption, we can obtain a meaningful discretization result. We will modify our assumption and add additional experiments in our next version paper. Thanks again for the detailed discussion.

---

### Meta-Review · Area_Chair_2f6q · 2024-12-15

**Metareview:**

This paper takes an important step in understanding the empirical success of newly proposed Consistency models. However, while the contributions are greatly appreciated, the rebuttal phase revealed that the paper appears to be relatively incomplete. It is missing critical derivations and discussions. Precise points are given in the reviews.

**Additional Comments On Reviewer Discussion:**

The reviewers requested clarifications on several derivations and also requested discussions. The authors largely addressed the concerns. The discussions reveal that the paper needs to be modified considerably.

---

### Decision · Program_Chairs · 2025-01-22

Reject